# Distinct subnetworks of the mouse anterior thalamic nuclei

Houri Hintiryan [1] ✉, Mitchell Rudd [1,7], Sumit Nanda[1,7], Adriana E. Gutierrez[1], Darrick Lo[1], Tyler Boesen[1], Luis Garcia[1], Jiandong Sun [1], Christian Estrada[1], Hyun-Seung Mun[1], Seita Yamashita[1], Yeji E. Han[1], Ian Bowman[1], Lin Gou[1], Chunru Cao[1], Jennifer Gonzalez[1], Keivan Moradi[1], Qiuying Zhao[1], Inga Yenokian[1,5], Aishwarya Dev[1], Brian Zingg[1], Hanpeng Xu[1], Qing Xue[1], Muye Zhu[1,6], Lijuan Liu [2,3], Xin Chen[2], Zhixi Yun[2], Hanchuan Peng [4], Nicholas N. Foster [1] & Hong-Wei Dong [1] ✉

Currently, classification of neuron types in the mouse thalamus remains largely incomplete. The anterior thalamic nuclei (ATN), a Papez circuit component, encompass the anterodorsal (AD), anteroventral (AV), and anteromedial (AM) thalamic nuclei. Structurally, the ATN facilitate communication among the neocortex, hippocampus, amygdala, and hypothalamus. Functionally, they play pivotal roles in learning, memory, spatial navigation, and goal-directed behaviors. Therefore, the ATN provide a promising avenue to investigate the relationship between structural and functional complexity with neuron type diversity. In male mice, comprehensive, systematically collected, pathway tracing data revealed several connectionally unique ATN cell populations, suggesting multiple parallel subnetworks run through each nucleus. Further, we applied genetic sparse labeling, brain clearing, 3D microscopic imaging, and computational informatics to morphologically characterize and catalog ATN neuron types. This study provides insights into how the prefrontal cortex, hippocampus, and amygdala interact through neuron type-specific ATN subnetworks to coordinate cognitive and emotional aspects of goal-directed behavior.

The Papez circuit, comprising the hippocampal formation, mammillary bodies, anterior thalamic nuclei (ATN), cingulate cortex, and parahippocampal cortices, was proposed as a foundation for emotional expression[1]. Building upon this, the concept of the limbic system or visceral brain was introduced, which incorporated additional structures like the amygdala, hypothalamus, and septum. Since then, this model has been revised to underscore the cognitive roles of these structures in episodic memory, attention, and spatial processes[2,3].

Within this structural framework, the ATN serve as a multifunctional hub, supporting diverse aspects of cognition in both animal models and humans[2].

The ATN consists of the anterodorsal (AD), anteroventral (AV), and anteromedial (AM) thalamic nuclei, the latter of which is further divided into dorsal (AMd) and ventral (AMv) compartments[4]. Together, the ATN support functions of memory and spatial navigation[5]. Pathology of the ATN culminates in diencephalic amnesia[6,7],

[1]UCLA Brain Research & Artificial Intelligence Nexus (B.R.A.I.N.), Department of Neurobiology, David Geffen School of Medicine at UCLA, University of California Los Angeles, Los Angeles, CA, USA. [2]New Cornerstone Science Laboratory, SEU-ALLEN Joint Center, Institute for Brain and Intelligence, Southeast University, Nanjing, China. [3]School of Biological Science & Medical Engineering, Southeast University, Nanjing, China. [4]Shanghai Academy of Natural Sciences (SANS), Fudan University, Shanghai, China. [5]Present address: Cedars-Sinai Medical Center, Los Angeles, CA, USA. [6]Present address: Google, Boulder, CO, USA. [7]These authors contributed equally: Mitchell Rudd, Sumit Nanda. ✉e-mail: HHintiryan@mednet.ucla.edu; HongWeiD@mednet.ucla.edu

inactivation of the nuclei manifests in perturbations of spatial functions[8–10], and elevated ATN activity rescues spatial memory loss sustained following mammillothalamic tract insult[11]. Increasing evidence also suggests the ATN's contribution to age-related cognitive changes[12]. Further, many hippocampal-dependent memory processes are shown to require a functional ATN, emphasizing its role in the hippocampal-diencephalic-cingulate network[9,13,14].

Based on their unique connections, three parallel hippocampal-ATN streams have been suggested[15]. The AD is implicated in head directionality because it harbors the largest number of head direction cells within the thalamus[16–19] and is nestled in a network whose components exhibit head direction specificity like the lateral mammillary nucleus (LM), dorsal tegmental nucleus (DTN), and the postsubiculum (POST) (DTN→LM→AD↔POST)[17,20]. The AV exhibits theta rhythm oscillations and is interconnected with structures also implicated in theta rhythmicity imperative for spatial and non-spatial mnemonic functions[21–23]. These structures include the medial mammillary nucleus (lateral part) (MMl), ventral tegmental nucleus (VTN), and the subiculum (SUB) (VTN→MMl→AV↔SUB)[20]. More recent studies have shown that through their connections with the retrosplenial cortical area (RSP), the AD and AV are involved in contextual memory and memory specificity, respectively[24]. Importantly, the AV is implicated in age-related decline in working memory that can be ameliorated through AV activation[25]. Alternatively, the AM has been proposed to serve as a transmitter of integrated hippocampal-diencephalic information to the cortex for higher order processing[15,26] and in goal-directed behaviors[27] given its unique connections with medial prefrontal cortical areas like the prelimbic (PL) and infralimbic (ILA) areas[28]. The AM in fact regulates emotional learning since its manipulation affects contextual fear responses[29–31], suggesting more than a relay role for the AM[32].

These studies stress the importance of attaining an intricate and complete connectivity roadmap of the ATN to reveal more granular, potentially functionally disparate, subnetworks. Although studies in rats have shown some segregation within the AM[33,34] and AV[35], a systematic and extensive examination of ATN connectivity that can reveal the exceedingly granular subnetworks to account for ATN functional diversity is lacking. To achieve this, we traced ~200 sets of cortical, thalamic, hippocampal, amygdalar, striatal, midbrain, and hindbrain pathways in mice, constructing a comprehensive ATN wiring diagram. We identified connectionally unique cell types (or domains) within the ATN, associated with seven parallel subnetworks within the AMd, four within the AV, and two within the AD, providing a structural basis for understanding the ATN's functional role as a network hub among the neocortex, hippocampus, amygdala, and hypothalamus. Combining genetic Golgi-like labeling, brain clearing, and 3D microscopic imaging with computational informatics, we systematically reconstructed and cataloged ATN neuron types based on their neuron morphology. Comparable single-cell morphological assessments have not been systematically conducted for the ATN in any species. Finally, we validated synaptic connections of ATN neurons that facilitate communication among the medial prefrontal cortex, hippocampus, and amygdala.

## Results

### Characterizing ATN domain connectivity

To assess ATN connectivity, anterograde (PHAL, AAV) and retrograde (CTB, FG, AAVretro-Cre) pathway tracers were placed across different cortical, hippocampal, amygdalar, thalamic, striatal, midbrain, and hindbrain areas (see Methods for details regarding each of these approaches and Discussion section, *Caveats to tracing experiments*, regarding unique tracing and neurotropic characteristics of the tracers). To determine the structure and approximate boundaries of domains (i.e., uniquely connected cell populations) within the AD, AV, and AM, representative experimental cases with distinct labeling in

each domain were selected and sections across the ATN were annotated and analyzed. The sections were processed through our proprietary software Outspector[36–39]. Individual sections were matched and warped to their corresponding atlas level of the Allen Reference Atlas (ARA[4]) and subsequently the tracer labels were segmented (Fig. 1a, b). Grid-based overlap annotation was performed in which the AD, AV, and AM were divided into 105×105-pixel square grids (equivalent to 63 μm²) to tabulate labeling within distinct domains of the ATN nuclei (Fig. 1b). The case specific annotations were then aggregated into a single matrix and Louvain community detection was conducted (Fig. 1c). Grids were color-coded according to community assignment and reordered such that the resulting Louvain clusters were placed along the diagonal of the visualized matrix (Fig. 1d).

To validate domain-specific connections, Cre-dependent anterograde and TVA receptor-mediated rabies tracing strategies were applied utilizing connectionally guided delivery of Cre. For these cases, sections spanning the entire brain were registered and tracer labels were segmented and annotated based on ROI. The annotated data underwent normalization and 2D hierarchical clustering to ascertain and visualize the distinct whole-brain projection patterns resulting from neurons traced in different ATN domains. A total of ~200 injections were used to generate connectivity diagrams and a global wiring diagram of the ATN (Fig. 1e; https://ucla-brain.github.io/atn/). Additional validation methods included repeated injections (Supplementary Table 1), retrograde tracers in regions displaying anterograde terminal labeling to confirm anterograde connections, anterograde tracers in regions with retrogradely-labeled projection cells to validate retrograde connections, and single neuron tracing experiments.

### General and laminar connections of the ATN

The AD, AV, and AM each have exclusive connections with the cortex and hippocampus[40] (Fig. 1f). Reciprocal communication between the medial prefrontal cortex (MPF) and ATN are specifically through the AM, projections from ATN to the post- (POST), pre- (PRE), and para (PAR) subiculum are exclusively through the AD and AV, while input from SUB is exclusively to AV, and output to the striatum is solely through the interanterodorsal thalamic nucleus (IAD; Fig. 1f), a ventral extension of the AD situated dorsal to the AMd. See Supplementary Table 2 for a list of structure abbreviations.

Regarding laminar specific connections, ATN thalamo-cortical neurons project to cortical layers I, IV, and V and to all subicular layers[34,41]. In turn, layer VI cortico-thalamic neurons and deeper subiculo-thalamic neurons[42,43] project back to ATN. Our data also show this canonical cortico-thalamic and thalamo-cortical patterns of labeling (Fig. 1g). An anterograde and retrograde tracer cocktail in the AM shows anterograde tracer labeling in MPF layers I, IV, and V, while layer VI neurons in the same regions are retrogradely labeled (Fig. 1g). A similar AV injection shows the same laminar specific connections with the ventral retrosplenial cortical area (RSPv). Projections from ATN to POST, PRE, and PAR spread across all three layers, while subicular neurons projecting back to ATN reside in deep layer III (Fig. 1g).

### AD subnetworks

Within the AD, we identified the AD.medial (AD.m) and AD.lateral (AD.l) domains (Fig. 2a). The AD projects to the POST, PRE, and PAR and to the dorsal (RSPd) and ventral (RSPv) retrosplenial cortical areas (Supplementary Fig. 1a). AD neurons projecting to POST, PRE, and PAR are located primarily in AD.medial (AD.medial→PRE/PAR/POST), while those projecting to RSPd/v predominate the AD.lateral (AD.lateral→RSPd/v) (Fig. 2b, c; Supplementary Fig. 1b−d). Clear segregation of AD.medial and AD.lateral neuronal populations is evident in a case in which retrograde tracers were placed in the RSPv and the PRE in the same animal (Fig. 2d). While AD.lateral connections with RSP are

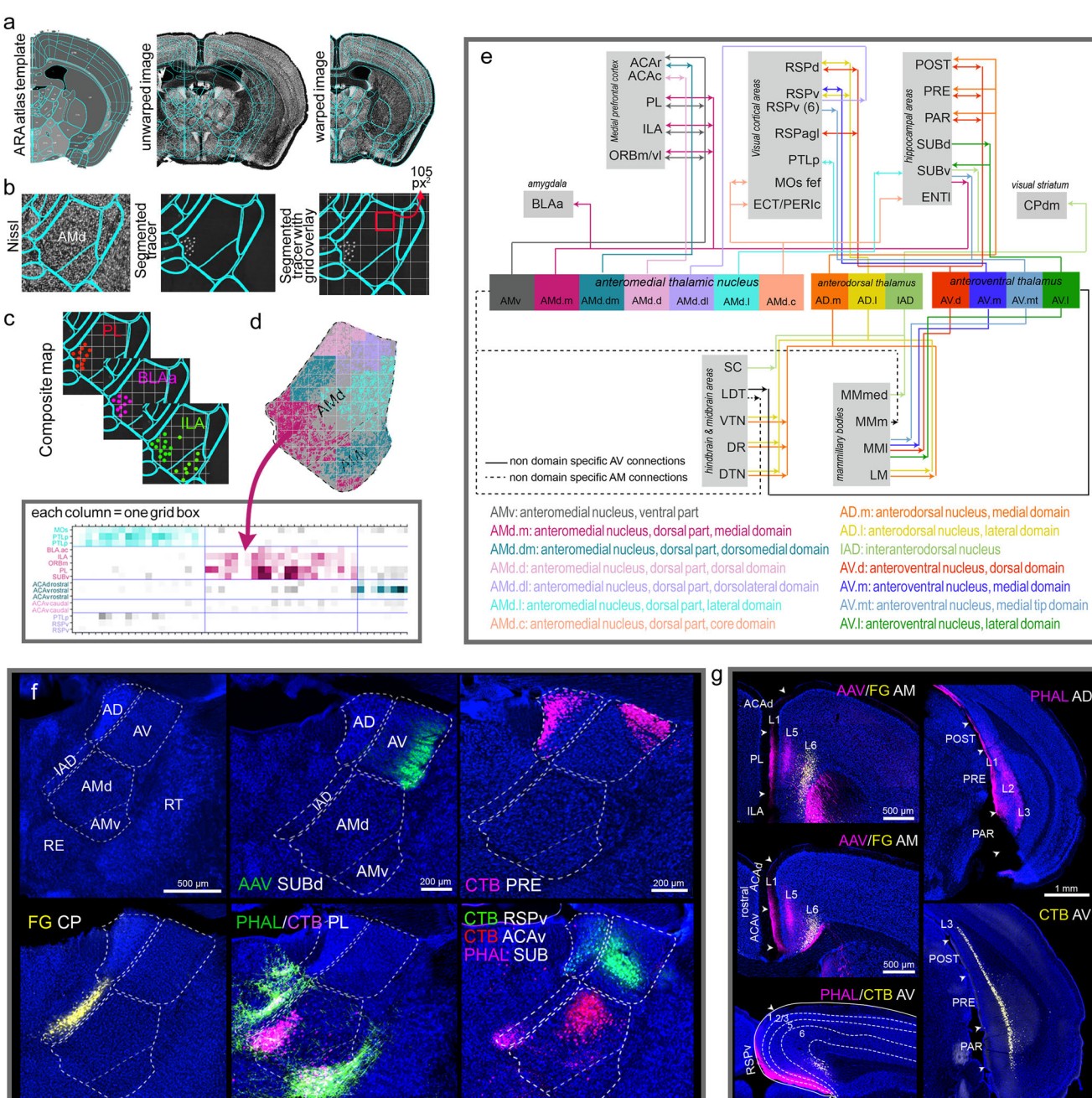

**Fig. 1 | Approach. a** Tissue sections from tracer injection cases are imported into Outspector, matched to their corresponding ARA atlas template level, and fiducial markers placed within the tissue Nissl channel are aligned with those in the atlas template. Tissue images are then deformably warped (e.g., registered) and (**b**) the tracer labeling is segmented (e.g., thresholded). The ATN is subdivided into 105 px$^2$ $^2$ grid cells and the axon (pixels) or cell (cell count) labeling within each grid square is quantified. **c** The aggregated quantified data from all injections to all the grid cells is then analyzed using a modularity maximization algorithm to group injection sites with terminations in common grid cells. The darker shading indicates stronger connections. **d** The output is visualized in a color-coded matrix to illustrate the subnetworks within each nucleus. **e** The generated data was used to create the whole-brain wiring diagram of the AD, AV, and AM and their identified

domains. **f** The overall organization of the ATN nuclei, followed by the unique connections of each nucleus, e.g., SUBd→AV, AD/AV→PRE, IAD→CP, PL↔AM, RSPv→AD/AV, and AM→ACACv. **g** The laminar organization of thalamo-cortical projections and cortico-thalamic projection neurons. An anterograde and retrograde tracer co-injection in the AM shows labeled thalamo-cortical projections to layers I–V and layer VI corticothalamic cells. The bottom left panel shows a similar laminar organization of thalamocortical projections from the AV. Right panels show laminar organization of hippocampal-thalamic projection neurons and thalamo-hippocampal projection terminals. A CTB retrograde tracer injection in the AV shows layer III labeled POST, PRE, and PAR projection neurons. An anterograde PHAL injection in the AD shows projections to layers I-III of POST, PRE, and PAR. See Supplementary Table 2 for list of structure abbreviations.

reciprocal (AD.lateral↔RSPd/v) (Fig. 2c; Supplementary Fig. 1b), AD.medial→POST/PRE/PAR connections are unidirectional (Fig. 2e). This unidirectionality is also demonstrated with an anterograde and retrograde tracer co-injection in the PRE which results in retrogradely labeled neurons, but not anterogradely labeled axonal terminals, in the

AD.medial (Fig. 2f). No connection between the AD and SUB proper were detected (Fig. 2g, h). Instead, those subicular connections were with AV (Fig. 2g, h; see next section for details). This data updates the canonical AD circuits that connect with the RSP and subiculum (Fig. 2i).

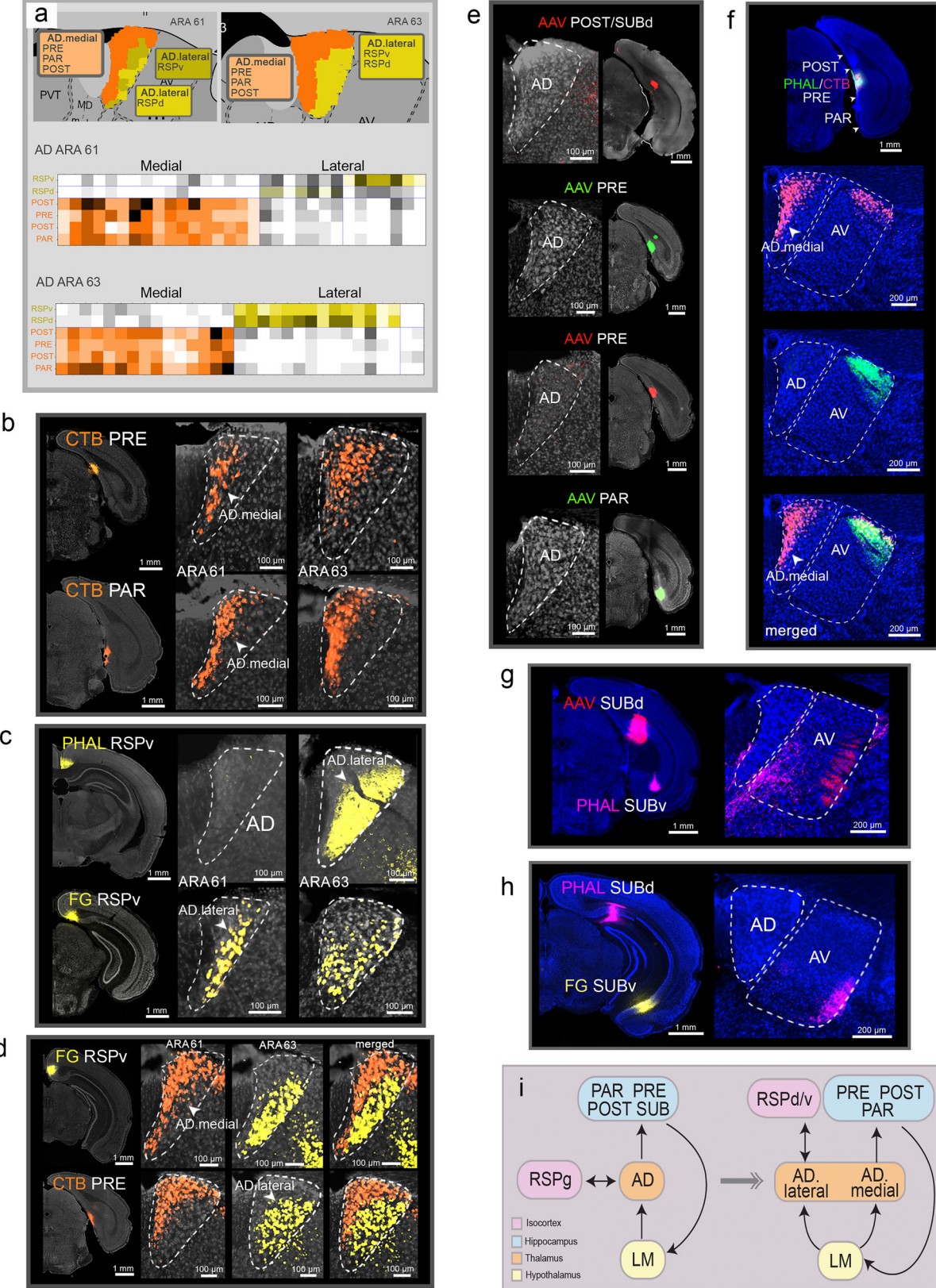

## AV subnetworks

Four domains with distinct connectional properties were identified within the AV. These were the AV.dorsal (AV.d), AV.lateral (AV.l), AV.medial (AV.m), and AV.medial tip (AV.mt) (Fig. 3a). The AV shares strong reciprocal connections with the POST, PRE, and PAR (Fig. 3b1, c1; Supplementary Fig. 2a, b). These connections are specifically with AV.dorsal (AV.dorsal↔POST/PRE/PAR) since anterograde and retrograde tracers in the POST, PRE, and PAR label only this domain (Fig. 3b2, c2; Supplementary Fig. 2b–d). Note the labeling in the mammillary bodies and the lateral dorsal thalamic nucleus supporting the precise location of ATN and subicular injections (Fig. 3b). The AV is also reciprocally connected with the RSPd, RSPagl,

**Fig. 2 | Connectivity of the AD. a** Based on network analysis of their connectivity, the AD was subdivided into two domains: the AD.medial and AD.lateral, across ARA 61 and 63. The matrices represent the percentage of a specified grid covered by axons or cells from each injection site. These data were subject to a modularity maximization algorithm that grouped injection sites (rows) with labels in common grid cells (columns) within the AD. The matrix was reordered to present the identified groups (i.e., domains) along the diagonal. Domain names are listed above the matrices. Finally, each AD pixel grid was recolored according to their community structure to visualize the domains on the atlas AD. A total of 6 cases (6 sections with AD labels at ARA 61 and another 6 sections at ARA 63) were included in the analysis. Injection ROIs are indicated in the rows of the matrix. **b** CTB retrograde tracer injections in the PRE and PAR show labeled neurons specifically in the AD.medial at ARA levels 61 and 63. **c** Anterograde (PHAL) and retrograde (FG) injections in the RSPv label mostly the AD.lateral at ARA 61 and 63. **d** Double retrograde injections in the PRE and RSPv in the same animal clearly demonstrate the AD.medial and AD.lateral distinction. **e** Numerous anterograde tracer injections placed across the POST, PRE, and PAR did not label axons in the AD. **f** AD.medial and AV.dorsal project to PRE (AD.medial/AV.dorsal→PRE), but PRE projects back only to AV (AV←PRE). A co-injection of an anterograde and retrograde into the PRE retrogradely labels AD.medial and AV.dorsal neurons, but anterogradely labeled terminals are present only in the AV.dorsal. **g** No connections were detected between the SUB and the AD. Anterograde tracers in the SUBd and SUBv do not label terminals in the AD. **h** Anterograde and retrograde injections show no labeling in the AD. **i.** Left is a canonical schematic of connections among the SUB, AD, RSP, and LM with the AD. Right is an updated version of those connections with the medial and lateral subnetworks running through the AD.

and RSPv (Supplementary Fig. 2e). The connections with the RSPd and RSPagl are through the AV.dorsal domain (AV.dorsal↔RSPd/agl; Fig. 3d), while those with RSPv are with the AV.medial domain (AV.medial↔RSPv; Fig. 3e; Supplementary Fig. 2f). Multiple tracer injections clearly demonstrate the AV.dorsal and AV.medial and their approximate boundaries (Fig. 3f). Their unique connections were also validated. When retrograde tracer injections were made primarily in either the AV.dorsal or AV.medial, labeled neurons in the POST/PRE/PAR and RSPd/agl were observed only with the AV.dorsal CTB injection (Fig. 3g). A different AV domain, the AV.medial tip, receives input from the SUBv and deeper layers of RSPv (SUBv/RSPv_6→AV.medial tip; Fig. 3h, i), which is validated with a retrograde injection in the AV.medial tip (Fig. 3g).

Finally, the AV.lateral is the domain where inputs from the SUBd predominantly target (SUBd→AV.lateral; Fig. 3j; Supplementary Fig. 2g). Double anterograde injections in the SUBd and SUBv in the same animal validate these connections and show the distinction between the AV.medial tip and AV.lateral (Fig. 3k; Supplementary Fig. 2h, i). AV neurons that project back to the SUB were difficult to reveal given the sparse AV→SUB projections to a very specific region of the caudal SUBv (Fig. 3l1). Many retrograde tracer injections made across the SUB did not label AV neurons. Only one in the caudal SUBv labeled neurons in the AV.lateral, highlighting the specific AV→SUB connection (Fig. 3l2). All the connections of the AV domains are summarized in a wiring diagram (Fig. 3m) and the unique connections of the ATN with the SUB are presented (Fig. 3n).

## AM subnetworks
Anterograde and retrograde tracers placed across different cortical, amygdalar, and hippocampal regions revealed segregated neuron populations within the AM and the connectional distinctions between the AMd and AMv (Fig. 4a). Computational analysis of the AMd connectivity data revealed 5 domains within the AMd (Fig. 4b, c), namely the medial (AMd.m), dorsomedial (AMd.dm), dorsal (AMd.d), dorsolateral (AMd.dl), and lateral (AMd.l).

The AM connects strongly with the dorsal (ACAd) and ventral (ACAv) anterior cingulate cortices (Fig. 4d). The AMd.dorsomedial domain (AMd.dm) contains neurons that project to the rostral parts of the ACAd and ACAv (AMd.dorsomedial→rostral ACAd/ACAv) (Fig. 4e; Supplementary Fig. 3a). The AMd.dorsomedial also receives input from the rostral ACAd/ACAv, but this rostral ACAv/ACAd→AMd.dorsomedial connection is far weaker (Fig. 4e). These AMd.dorsomedial connections were validated with repeated rostral ACA injections (Supplementary Fig. 3b1). The AMv also shares reciprocal connections with the rostral ACAd/ACAv (AMv↔rostral ACAd/ACAv) (Fig. 4e; Supplementary Fig. 3b1, c).

The AMd.dorsal (AMd.d) shares the strongest connections with the caudal ACAd/ACAv (AMd.dorsal↔caudal ACAd/ACAv) (Fig. 4f). These connections were validated with repeated caudal ACA injections (Supplementary Fig. 3b2), but also through a Cre-dependent anterograde AAV injection (see "Methods") that traced the output of

mostly AMd.dorsal neurons (Fig. 4g, h) that showed labeled terminals in caudal ACA, but none in rostral ACA. No connections were detected between the caudal ACAd/ACAv and the AMv (Fig. 4f; Supplementary Fig. 3b2). Multiple tracer injections made in the same animal highlight the distinct AMd.dorsomedial and AMd.dorsal domains (Fig. 4i; see Fig. 4j for a summary of AMd.dorsomedial and AMd.dorsal connections). Connections with the ACA in general are unique to the AM and no connection between the ACA and the AV nor AD were identified (Fig. 4k, l; Supplementary Fig. 3c).

Some connections from the AMd to the RSP were detected (Fig. 5a), which were specifically through the AMd.dorsolateral domain (AMd.dl) that houses the neurons that project to the RSPv (AMd.dorsolateral→RSPv) (Fig. 5b). This AMd.dorsolateral→RSPv connection was validated with repeated RSP injections (Supplementary Fig. 3d) and through a Cre-dependent anterograde injection that traced the output of primarily AMd.dorsolateral neurons (Fig. 5c, d). No projections back to this domain, nor to any other AMd domain, from the RSPv were identified (Fig. 5e). Similarly, no connections between the AMv and RSP were detected (Supplementary Fig. 3d).

The AMd is bidirectionally connected with the posterior parietal cortex (PTLp; Fig. 5f). The AMd.lateral domain (AMd.l) is shown to project to and receive input from the PTLp, specifically the caudal medial part (AMd.lateral↔PTLp caudal medial) (Fig. 5g; see Supplementary Fig. 3e for repeated PTLp injections). A Cre-dependent AAV anterograde injection that primarily traced AMd.lateral neurons shows projections to the PTLp (caudal, medial), while a similar injection in medial AMd shows no projections to the PTLp (Fig. 5h, i), validating this connectional specificity. The PTLp is not connected with AMv, the AD, nor AV (Fig. 5j). The AMd.lateral also receives input from the SUBv (AMd.lateral←SUBv) (Fig. 5k, l). Neurons in the AMd that project back to the SUB were difficult to find given the sparse AM→SUB projections and given that these projections are to a very specific region within the caudal SUBv (Supplementary Fig. 4a). A CTB retrograde tracer injection in this caudal SUBv region labeled neurons predominantly in the AMd.lateral (Supplementary Fig. 4b). Multiple tracer injections in the same brain further demonstrate the select AMd.lateral and AMd.dorsolateral domains (Fig. 5m).

The fifth AM domain is the AMd.medial (Fig. 6a). Although undetected through the modularity maximization algorithm, there were two distinct zones within the AMd.medial domain: (1) the AMd.medial tip (AMd.mt) and (2) AMd.ventromedial (AMd.vm) (Fig. 6a).

The AM shares strong connections with the MPF, SUBv, and amygdala as shown by anterograde and retrograde AM injections (Fig. 6b, d), which all occur either through the AMd.medial tip, AMd.ventromedial, or the AMv. The AD and AV are not connected with the MPF nor the amygdala (Fig. 1f). The AMd.medial tip receives inputs from SUBv and some light input from PL and ILA (SUBv→AMd.medial tip; PL/ILA→AMd.medial tip), but strongly projects to the PL (AMd.medial tip→PL) and to the anterior basolateral amygdala (AMd.medial tip→BLAa) (Fig. 6b–e; Supplementary Fig. 4c–e). These AMd.medial tip outputs were validated with a Cre-dependent anterograde AAV

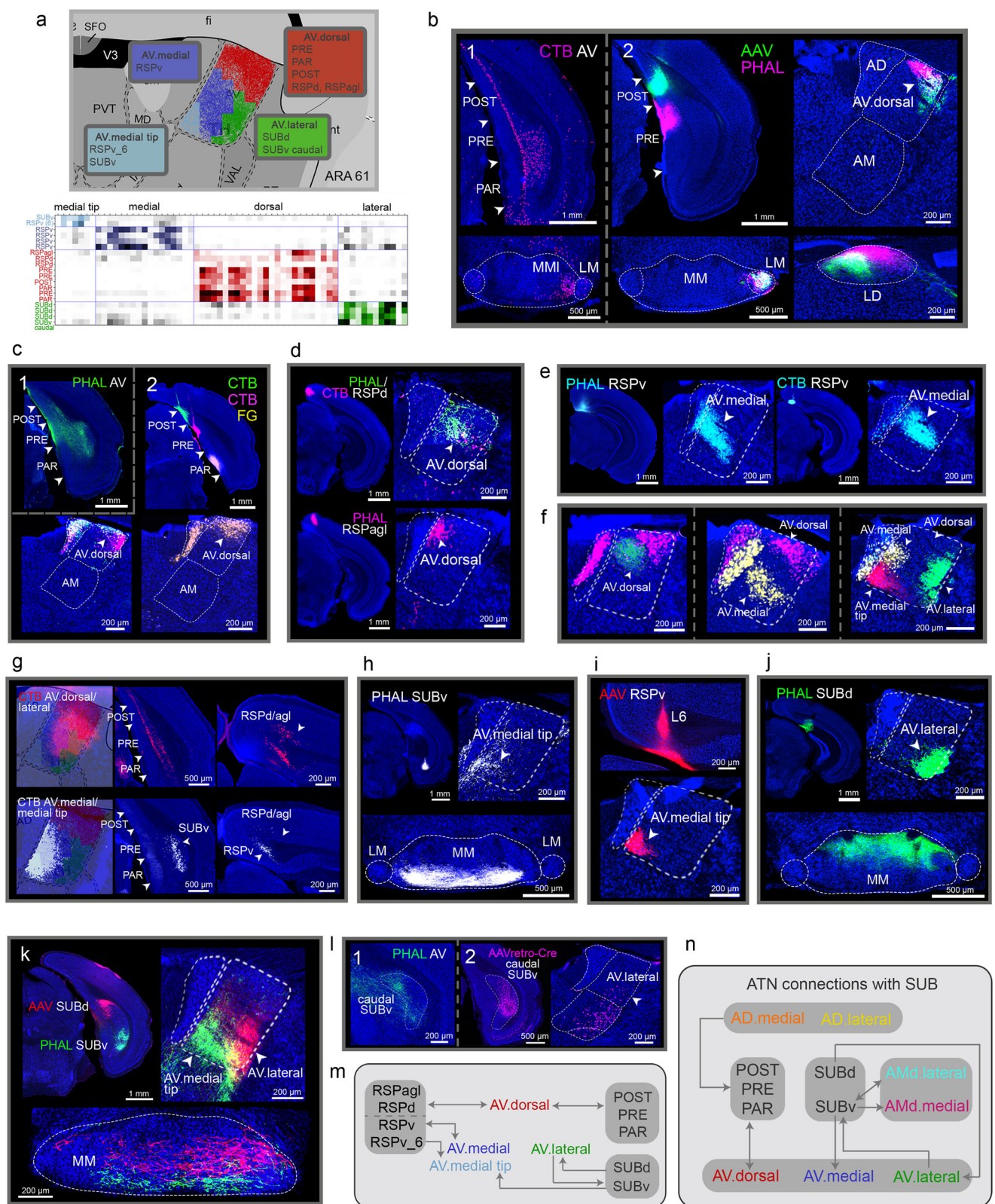

injection that primarily traced neurons in this domain (Fig. 6f), while their inputs were validated through TVA receptor-mediated rabies tracing (Supplementary Fig. 4f–h; see also Supplementary Fig. 4c–e for repeated injections).

The AMd.ventromedial domain also receives light input from PL and ILA (none from the SUB) (PL/ILA→AMd.ventromedial) (Fig. 6c), has reciprocal connections with the medial (ORBm) and ventrolateral (ORBvl)

ORB (AMd.ventromedial↔ORBm/vl) (Fig. 6g, h), and strongly projects back to ILA (AMd.ventromedial→ILA) (Fig. 6e; Supplementary Fig. 4e). Multiple injections made into the ILA and PL illustrate the distinct AMd.medial tip and AMd.ventromedial neuronal populations (Fig. 6i; Supplementary Fig. 4d, e) and quantification of the Cre-dependent tracing from each of these populations validates the stronger AMd.medial tip→PL and AMd.ventromedial→ILA connections (Fig. 6j).

**Fig. 3 | Connectivity of the AV. a** The AV is subdivided into 4 domains, AV.medial tip, AV.medial, AV.dorsal, and AV.lateral, based on connectivity. The matrix presents Louvain-identified AV domains and shows the injection sites that group together based on the similarity of axonal or cell labeling they produce at each AV grid box. Nineteen cases (19 sections at ARA 61) were included in the analysis (injections are listed in the rows). Repeated injections for a single ROI (e.g., SUBd) were included to show they group together. Top: Recolored AV grids visualize the domains, and regions that each domain connects with are listed. **b1** CTB in AV backlabels neurons in POST, PRE, PAR deep layers. MMI labeled cells verify the AV injection location. **b2** Anterograde tracers in POST and PRE (labeling in LM and LD verify injection location) show POST/PRE/PAR (layer III)→AV.dorsal. Note the absence of labels in AD, suggesting no POST, PRE, PAR inputs to AD. **c1** AV PHAL injection labels all layers of POST/PRE/PAR (AV→POST/PRE/PAR). **c2** Retrograde injections in POST, PRE, PAR in a single animal reveal this as AV.dorsal→POST/PRE/ PAR. **d** RSPd and RSPagl tracer injections show their connections with AV.dorsal (RSPv/RSPagl↔AV.dorsal). **e** RSPv injections show their connections with AV.lateral

(RSPv↔AV.lateral). **f** Multiple tracer injections illustrate AV domain distinctions. **g** Retrograde tracer injection in AV.dorsal/AV.lateral validates POST/PRE/PAR→A-V.dorsal and the RSPagl→AV.dorsal connections. A retrograde tracer primarily in AV.medial/medial tip labels SUBv (not POST/PRE/PAR) and RSPv (not RSPagl), corroborating the distinct SUBv/RSPv→AV.medial tip connections. **h** AV.medial tip domain receives inputs from SUBv (MM label confirms SUBv injection site) and **i** RSPv deep layers. **j** The SUBd provides input to AV.lateral. SUBd→MM projection verifies SUB injection location. **k** Cre-dependent AAV injections in SUBd (red) and SUBv (green; AAVretro-Cre in AV) show their projections to distinct AV subregions (SUBd→AV.lateral; SUBv→AV.medial tip). SUBd and SUBv projections to MM validate their injection locations[54]. **l1** PHAL AV injection labels caudal parts of SUBv (AV→caudal SUBv), while **l2** a caudal SUBv CTB injection validates the connection and demonstrates it to be with AV.lateral (AV.lateral→caudal SUBv). **m** Wiring diagram summarizing AV domain connections. **n** ATN-SUB wiring diagram. See Supplementary Table 2 for abbreviations.

The strongest input from the PL, ILA, and ORB to the AM are to the AMv (PL/ILA/ORB→AMv) (Fig. 6c; Supplementary Fig. 4d, e) and the AMv has strong projections back to the PL (AMv→PL) (Fig. 6e; Supplementary Fig. 4d, e). These AMv connections were validated with Cre-dependent AAV and TVA receptor-mediated rabies tracing (Supplementary Fig. 4g–i). Quantification of this data clearly shows the distinct AMv connections (Supplementary Fig. 4h, j). Schematic summaries of these connections are provided (Fig. 6k; Supplementary Fig. 4k).

A final AMd domain not identified through our computational analysis, but was apparent from the tracing data, was the AMd.core. All domains discussed thus far predominate the peripheral region of the AMd, generally leaving its core region unoccupied by label (Fig. 4a, b). This peripheral-core organization is most apparent with multiple tracer injections made in a single brain (Fig. 7a). The AMd.core shows reciprocal connections with the secondary motor cortex (MOs), specifically with the frontal eye field region (AMd.core↔MOs.fef) (ACAd adjacent[44]) (Fig. 7b–d). Cre-dependent AAV tracing of these AMd.core neurons confirms strong AMd.core→MOs-fef connections (Fig. 7e). The AMd.core also houses neurons that project to the lateral entorhinal cortex (AMd.core→ENTl) (Fig. 7f–h). The AMd.core→MOs.fef and ENTl connections are unique to the AMd (Supplementary Fig. 3c), and the AMd.core→ENTl projections are to a very specific region of the ENTl layer V (Fig. 7f, g). No connections with the medial entorhinal cortex (ENTm) were detected (Fig. 7f, g). Exclusive projections from the AMd.core were also observed to the deep layers (V/VI) of the caudal ectorhinal and perirhinal cortical areas (AMd.core→ECT/PERI caudal) (Fig. 7f, g, i).

The data from the domain-specific Cre-dependent anterograde tracing from the AMd.dorsal/dorsolateral, AMd.ventromedial, AMd.medial tip, AMd.core, and AMv were grouped and subjected to a 2D hierarchical clustering algorithm to show the specificity of connections between these domains with the (1) MOs, (2) ORB, (3) PL, (4) ILA, (5) RSP, and (6) PTLp (Fig. 7j). The output showed that neurons that project most strongly to the PL are in the AMd.medial tip, while those that project most strongly to the ILA are in the AMd.ventromedial domain, validating our proposed domain model. The output also shows the projections from the AMv to the PL, ILA, ORBm, ORBvl, and shows no projections from AMv to the ORBl. Note also the projections from the AMd.core to the MOs and from the AMd.dorsolateral to the RSPv (Fig. 7j). Similar patterns emerge when data from injections administered in the broad AMd medial area compared to those in the AMd lateral area are analyzed (Fig. 7k), showing a clear distinction between the AMd ventral-medial and dorsal-lateral domains, with the medial domains connected more with limbic network structures and the lateral domains with visual processing network areas (Fig. 7l).

Axon terminations of singly traced neurons in the AMd.medial, AMd.dorsal/dorsolateral, and AMd.core also support the domain-

specific tracing results (Fig. 7m). A singly traced neuron from the AMd.medial showed greater axonal terminals in PL and ILA than neurons located more dorsally/dorsolaterally in the AMd. On the other hand, a neuron located more dorsal/dorsolaterally in the AMd showed more axonal terminations in the RSPv compared to the neurons located more medially and in the core region. Finally, only the neuron located in the AMd.core showed axon terminations in the MOs compared to the neurons located more in the medial and dorsal/dorsolateral parts of the AMd (Fig. 7m).

## AMd.medial and AMv serve as subnetwork hubs for subiculum and medial prefrontal cortical areas

The AM domains exclusively connect with the MPF, SUBv, and BLAa, indicating they serve as network hubs that facilitate communication across the cortex, hippocampus, and amygdala, which are structures involved in memory formation. The AMd.medial tip domain receives input from the SUBv, and in turn, projects to the PL and BLAa, suggesting that the AMd.medial tip regulates communication between (1) the SUBv and the BLAa via a SUBv→AMd.medial tip→BLAa circuit and (2) the SUBv and PL through a SUBv→AMd.medial tip→PL circuit.

We utilized an AAV1-Cre-based transsynaptic circuit mapping method[45] combined with a genetic sparse labeling reporter line MORF3[46] to validate these disynaptic circuits. In MORF3 mice, neurons stochastically and Cre-dependently express a membrane-targeted V5 spaghetti monster protein, which can be visualized via immunostaining. This method reveals the complete, detailed neuronal morphology of MORF3-labeled neurons, including dendritic arborizations and spines. In MORF3 mice, an AAV viral tracer expressing synaptophysin tagged with RFP was injected into the SUBv, and an AAVretro-Cre injection was made into the BLAa (Fig. 8a). In this strategy, Cre is retrogradely transported from the injection site to the AMd.medial tip to trigger MORF3 expression, consequently revealing the detailed dendritic morphology of the AMd.medial tip neurons in their entirety. Meanwhile, the synaptophysin-tagged AAV labels the synaptic terminals of fibers originating in the SUBv and terminating in the AMd.medial tip (Fig. 8b–d). As such, this method reveals SUBv fibers potentially synapsing onto BLA aprojecting AMd.medial tip neurons (Fig. 8e). The same strategy was applied to examine whether SUBv fibers form synaptic contacts onto PL projecting AMd.medial tip neurons (Fig. 8h-l). In both cases, putative contacts can be clearly seen from the SUBv onto BLAa and PL projecting neurons in the AMd.medial tip via the close apposition of terminals and dendrites (Fig. 8e, l). Synapse reconstructions further validated some of the potential contacts, substantiating the disynaptic circuits (Fig. 8f, g, m). Similarly, Fig. 8n, o illustrates the disynaptic ILA→AMv→PL circuit. Altogether, these data suggest that these AM domains serve as network hubs to bridge communication among the MPF, ventral hippocampus, and amygdala.

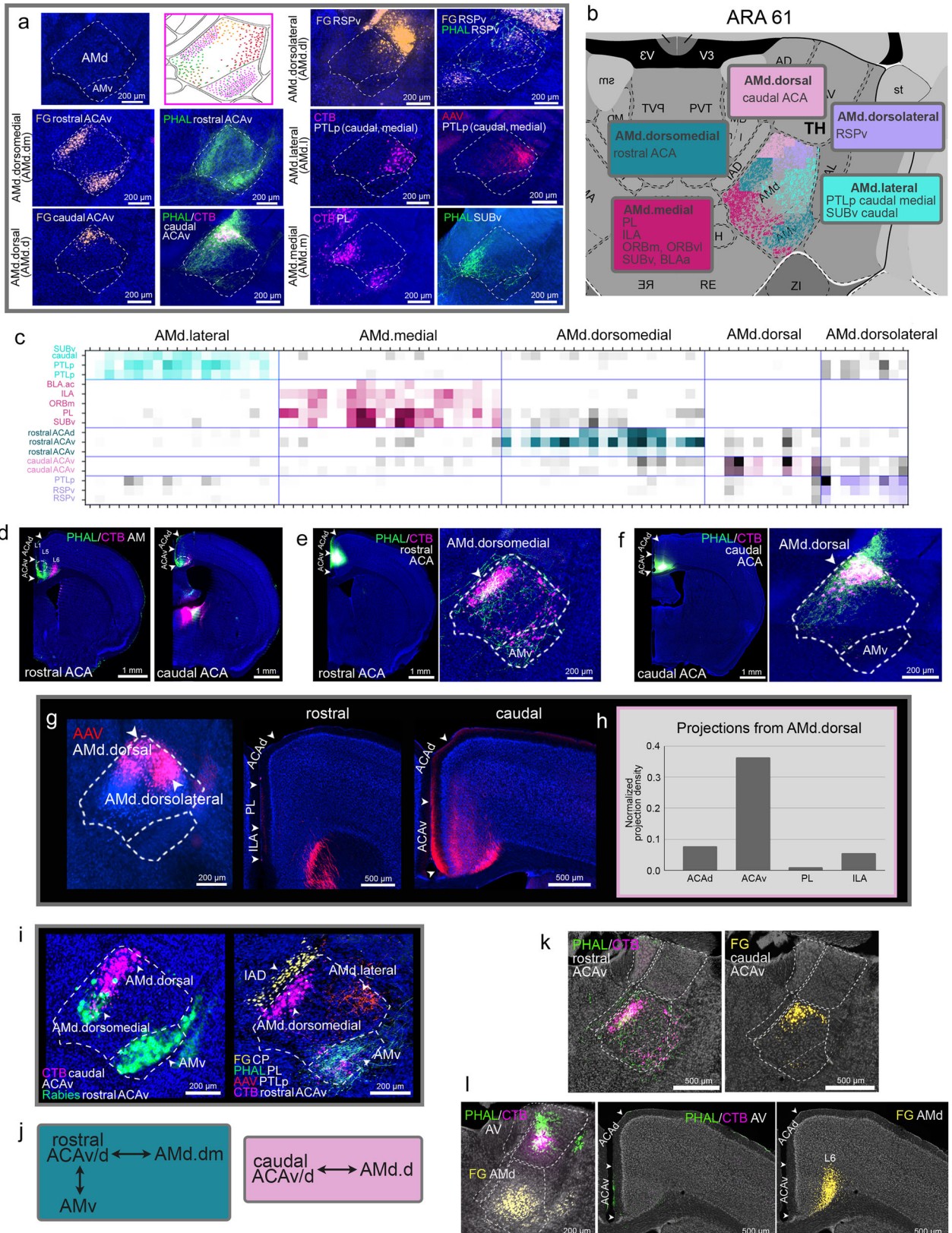

**ATN connections with the mammillary body and brainstem structures**

The ATN receive inputs from the mammillary nuclei in a topographically arranged manner although they appear non-specific with respect to the identified domains. The lateral mammillary nucleus (LM) projects to the AD, the lateral part of the medial mammillary nucleus

(MMl) to the AV, and the medial part of the MM (MMm) to the AMd and AMv (Fig. 9a, b; Supplementary Fig. 5a). The median MM (MMme) targets specifically the IAD (Fig. 9a1). Projections from the LM and MMme to the AD and IAD are bilateral, whereas those from the MMm to the AV or AM are ipsilateral (Fig. 9c, d). Cre-dependent TVA receptor-mediated rabies tracing validated these connections and

**Fig. 4 | Connectivity of the AMd.dorsomedial and AMd.dorsal domains. a** Tracer injections across cortex and hippocampus label different AMd domains (ARA 61). Manually mapped tracer labels highlight their distinct locations in the AM. **b** AM grids recolored to visualize the domains identified by the Louvain are shown, and the ROIs that each domain connects with are listed. Note the common connections of AMd.medial and AMv with MPF areas, also evident in the raw data. **c** The matrix presents the identified Louvain groups (i.e., domains) and shows the injection sites that are grouped together based on the similarity of labeling they produce at each AM grid box. Sixteen cases (16 sections with AM labeling at ARA 61) were included in the analysis. Injection ROIs are indicated in the rows of the matrix. **d** A PHAL/CTB AM co-injection labels cells (layer VI) and terminals (layers I-V) in rostral (left) and caudal (right) ACAv/ACAd, suggesting rostral/caudal ACA↔AMd connections. Dashed circles denote location of co-injections in (**e**, **f**). **e** A PHAL/CTB co-injection made in rostral ACA labels terminals and cells specifically in the AMd.dorsomedial domain (rostral ACA↔AMd.dorsomedial) and AMv (rostral ACA↔AMv). **f** A PHAL/

CTB co-injection made in caudal ACA labels terminals and cells specifically in the AMd.dorsal domain (rostral ACA↔AMd.dorsal) and does not label AMv neurons. **g** A Cre-dependent AAV injection that traces mostly AMd.dorsal (+dorsolateral) neurons (AAVretro-Cre injection made in ACA/RSP) shows projections only in caudal ACA, confirming caudal ACA↔AMd.dorsal connections. **h** Quantified projections to ACAd, ACAv, PL, and ILA from traced AMd.dorsal neurons in (**g**), validate their stronger projections to ACAv compared to other ROIs like PL and ILA. **i** Multiple tracer injections across different ROIs in the same brain showcase the distinct AMd domains. **j** Summarized connections of the AMd.dorsomedial (left) and AMd.dorsal (right). **k** Rostral and caudal ACAv injections show that ACA connectivity is selectively with AM, and not AV nor AD. **l** This selective AM-ACA connection is also shown with tracer injections made in the AV and AMd of the same brain. Neither anterograde nor retrograde AV injections produce labels in ACA, while AMd retrograde injection labels neurons in ACA deep layers.

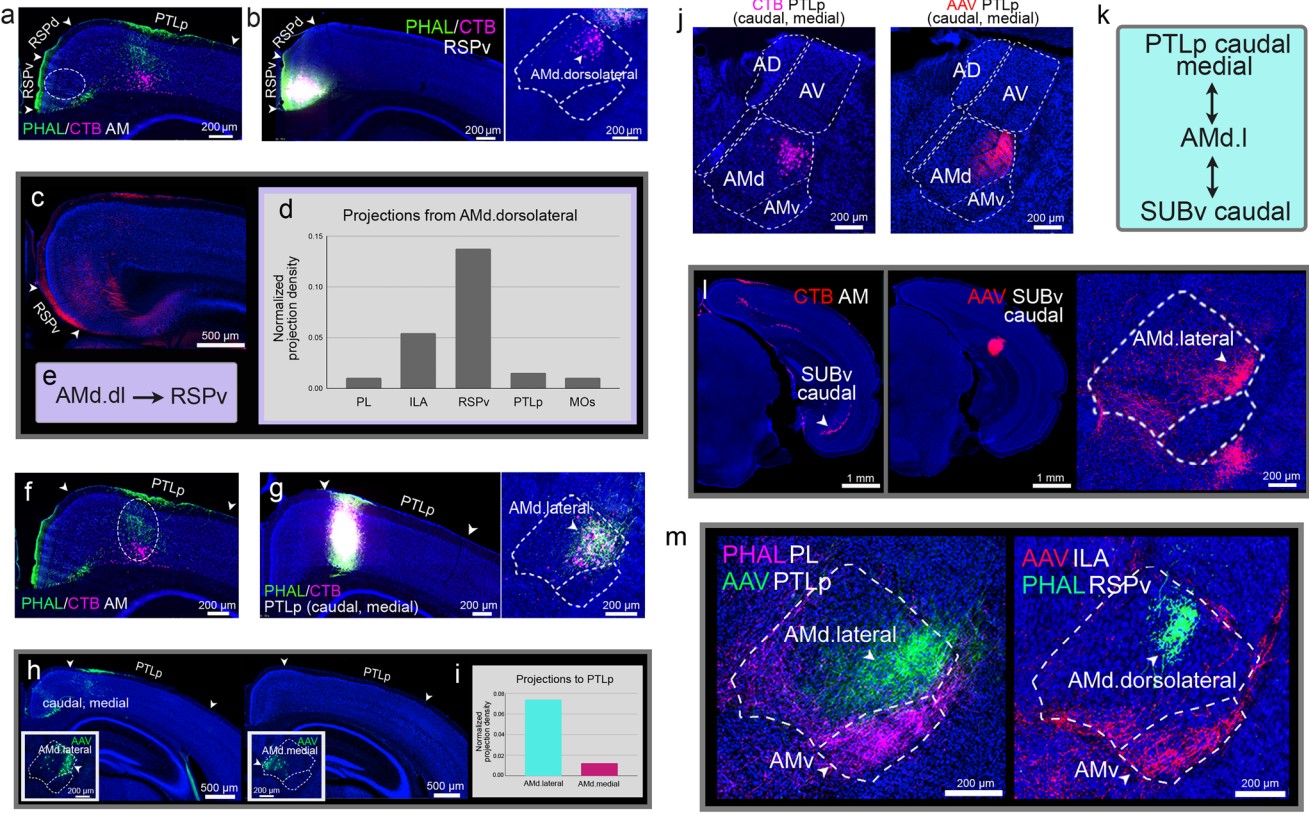

**Fig. 5 | Connectivity of the AMd.dorsolateral and AMd.lateral domains. a** A PHAL/CTB AM injection labels only fibers in the RSPv. Dashed circle denotes the location of a PHAL/CTB co-injection in the RSPv in (**b**). **b** PHAL/CTB co-injection back labels cells in the AMd.dorsolateral domain (AMd.dorsolateral→RSPv). Note the absence of fibers in the AMv. **c** A Cre-dependent anterograde injection that tracers AMd.dorsolateral neurons labels the RSPv. **d** Quantified projections to the PL, ILA, RSPv, PTLp, and MOs from these traced neurons in the AMd.dorsolateral domain show strongest projections to the RSPv compared to other regions. **e** Wiring diagram of the AMd.dorsolateral domain. **f** A PHAL/CTB anterograde and retrograde co-injection labels terminals and cells in the PTLp. Dashed circle denotes location of co-injection in (**g**). **g** A PHAL/CTB injection in the PTLp (caudal, medial part) labels terminals and cells in the AMd.lateral domain. **h** Cre-dependent

AAV injections made in the AMd.medial tip (left: AAVretro-Cre in PL/ILA) and in the AMd.lateral (right; AAVretro-Cre in PTLp). Only traced neurons in the AMd.lateral label the PTLp, specifically the caudal medial part (AMd.lateral↔PTLp caudal medial). **i** Bar graph shows the quantification of these projections to the PTLp, which are greater from traced AMd.lateral neurons versus those traced from the AMd.medial. **j** Connections with the PTLp are exclusively through the AM. **k** Wiring diagram of the AMd.lateral connections. **l** An AM CTB injection labels SUBv (caudal) cells. Dashed circle denotes location of an AAV injection made in the same SUBv caudal region that labels terminals in AMd.lateral (SUBv→AMd.lateral). **m** Multiple tracer injections in a single brain show the distinct AMd.lateral and AMd.dorsolateral domains. See Supplementary Table 2 for structure abbreviations.

demonstrated the AMd and AMv disynaptic connections (Fig. 9e). Specifically, a large AAVretro-Cre MPF injection was made that retrogradely delivered Cre to the AM. The AMd or AMv was then injected with a Cre-dependent TVA helper and EnvA-pseudotyped rabies viruses to trace their monosynaptic inputs (Fig. 9e). Dense rabies labeling is shown in the MMm, confirming disynaptic MMm→AMd/AMv→MPF pathways (Fig. 9e).

It is well established that the AD is embedded within a network processing head directionality, including the dorsal tegmental nucleus (DTN) pathway: DTN→LM→AD↔POST/PRE/PAR. In contrast, the AV is part of a network involved in theta rhythmicity and the VTN: VTN→MMl→AV↔SUB. Anterograde tracer injections in the DTN and VTN confirmed their established connections with the LM (DTN→LM) and MM (VTN→MMl), while also revealing unexpected direct

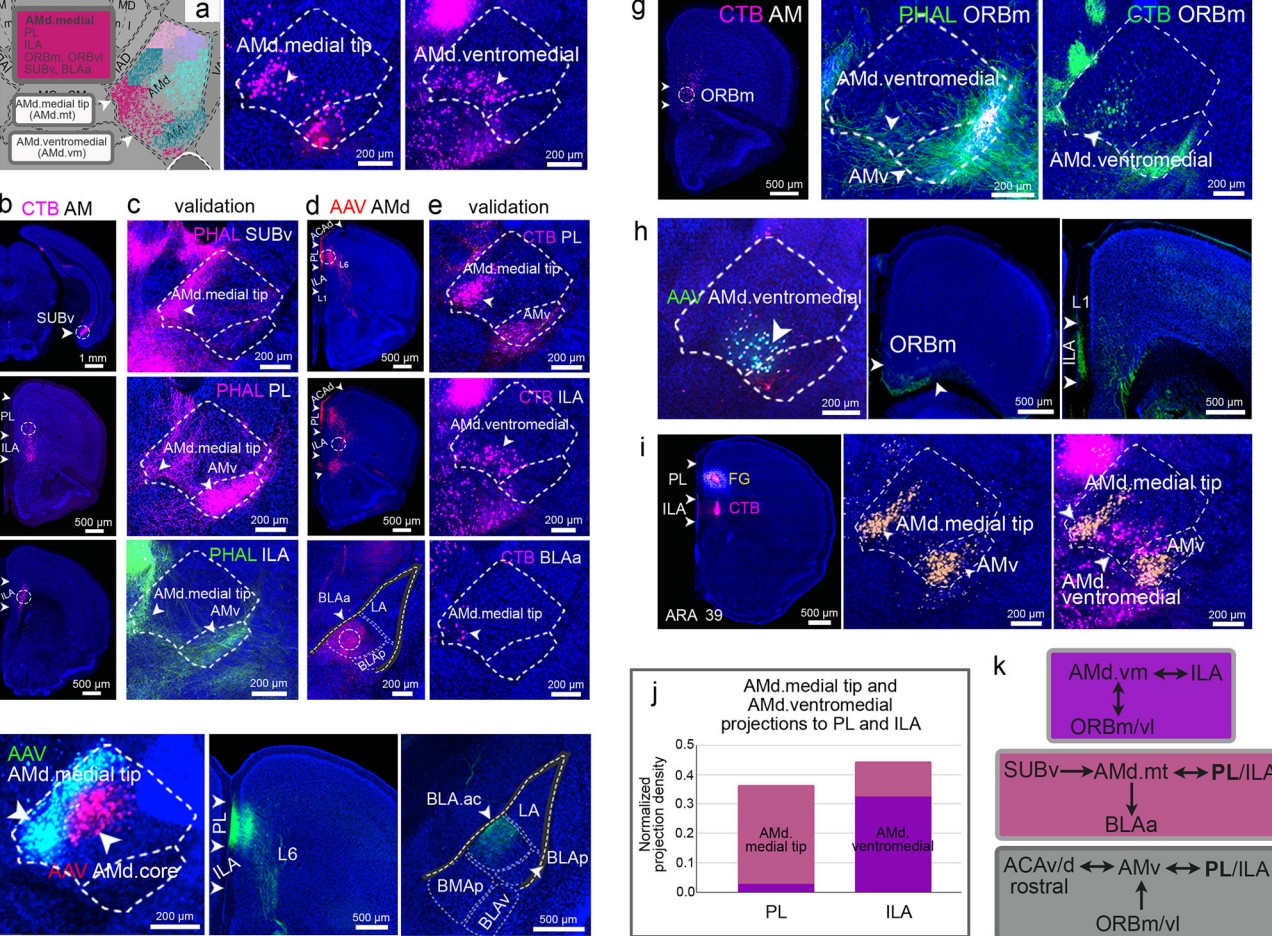

**Fig. 6 | Connectivity of the AMd.medial tip and AMd.ventromedial domains.**
**a** The AMd.medial can be divided into the AMd.medial tip and AMd.ventromedial domains, each of which shows distinct connections. **b** An AM CTB injection labels cell in the SUBv (top), PL (middle), and ILA (bottom). Dashed circles denote locations of PHAL injections in (**c**). **c** PHAL injections in the SUBv (top), PL (middle), and ILA (bottom) show their distinct connections with the AMd.medial tip and AMv (SUBv/PL/ILA→AMd.medial tip; PL/ILA→AMv). **d** An AM AAV injection labels fibers in the PL (top), ILA (middle), and BLA (bottom). Dashed circles denote location of retrograde tracer injections in (**e**). **e** Retrograde tracer injections in the PL (top), ILA (middle), and BLA (bottom) show their connections with the AMd.medial tip, AMd.ventromedial, and the AMv (AMd.medial tip/AMv→PL; AMd.medial tip/ventromedial/AMv→ILA; AMd.medial tip→BLA). **f** Cre-dependent AAV injections traced AMd.medial tip neurons (green; AAVretro-Cre in MPF) or avoided those neurons and traced only those in the AMd.core (red; AAVretro-Cre in ACA/MOs). Only the AMd.medial tip traced neurons labeled terminals in the PL, ILA, and BLA validating those connections showing the specificity of the domain-level

connections. **g** A CTB AM injection (left) labels neurons in the ORBm. Dashed circle denotes location of the PHAL and CTB ORBm injections in the right, which show labeled fibers and cells in the AMd.ventromedial domain (AMd.ventromedial↔ORBm). **h** A Cre-dependent AAV injection that traces primarily AMd.ventromedial neurons (AAVretro-Cre in the MPF) labels terminals in the ORBm and ILA validating those projections. **i** Double retrograde tracer injections in the PL and ILA in the same animal show (1) the distinct AMd.medial tip and AMd.ventromedial domains, (2) the select AMd.ventromedial/AMv→PL connections, and (3) less selective AMd.medial tip/AMd.ventromedial/AMv→ILA projections. **j** Quantification of projections to the PL and ILA from the Cre-dependent AAV injections that traced neurons in either the AMd.medial tip or the AMd.ventromedial validates the stronger AMd.medial tip→PL and the AMd.ventromedial→ILA connections. **k** Wiring diagrams summarizing the connections of the AMd.ventromedial (top), AMd.medial tip (middle), and AMv (bottom) domains. See Supplementary Table 2 for structure abbreviations.

connections from the VTN and DTN to the AD/AV (Fig. 9f, g), which were validated through retrograde tracing (Fig. 9h, i). These findings summarized in Fig. 9j provide insights into a potential crosstalk between two relatively segregated visuospatial and theta rhythmicity subnetworks (Fig. 9k).

We also identified a less documented direct projection from the dorsal raphe nucleus (DR) to the AD and AV (DR→AD/AV), providing serotonergic input to the ATN (Supplementary Fig. 5b, c). Combining CTB AD/AV retrograde tracing and immunostaining, we confirmed that the majority of AD/AV projecting DR neurons co-localize with tryptophan hydroxylase 2 (TPH2), a protein marker of serotonigic neurons (Supplementary Fig. 5d). The AV and AM, but not AD, also receive cholinergic inputs from the laterodorsal tegmental nucleus (LDT→AV/AM) (Supplementary Fig. 5e, f).

Lastly, our tracing revealed an unreported neural network involving the IAD. AMd anterograde tracer injections occasionally labeled terminals in the dorsomedial caudoputamen (CP) (Supplementary Fig. 6a, b). Retrograde tracer injections into the dorsomedial CP labeled neurons exclusively in the IAD, not the AM (Supplementary Fig. 6a–c), indicating strong IAD→CP connections. This was further validated by Cre-dependent anterograde tracers injected into the AMd: one involving the IAD and one excluding it. Only the injection that traced IAD neurons labeled the dorsomedial CP (Supplementary Fig. 6d, e). These IAD→CP projections are to specific CP domains that integrate visual and spatial information[39] (Supplementary Fig. 6f). Sparse connections also were observed between the superior colliculus and the IAD (SC→IAD) (Supplementary Fig. 6g). Since the IAD also receives inputs from MMme

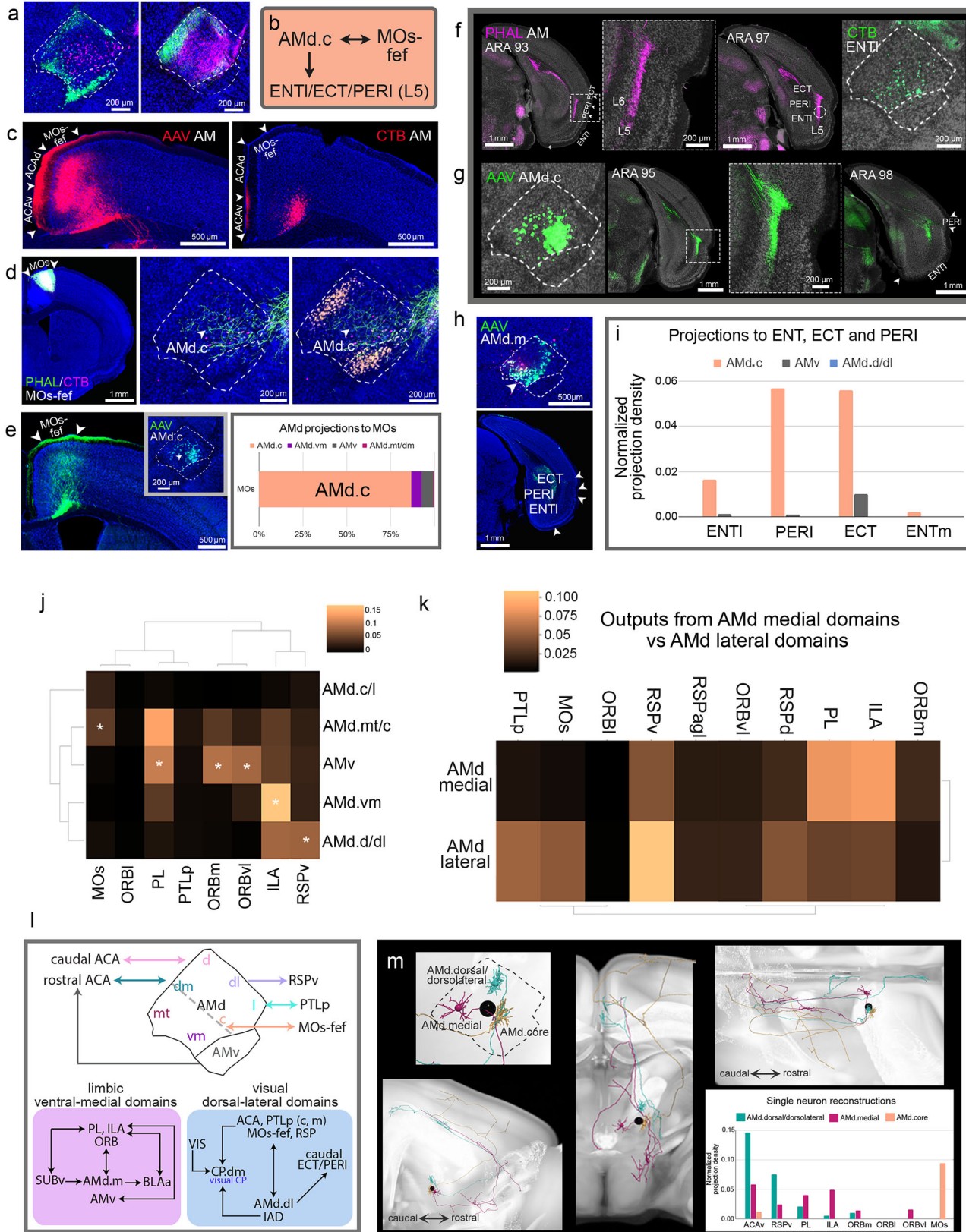

(Fig. 9a4), these data together suggest that the IAD transmits spatial (MM) and visual (SC) information to the dorsomedial CP (MMme/SC→IAD→CPi.dm/CPc.d), which also receives direct inputs from visual, spatial, entorhinal cortices implicated in coordinating eye, head, and neck movement for goal-directed behavior[39] (Supplementary Fig. 6h).

## ATN dendritic morphology

Dendrites are crucial in how neurons receive and integrate signals from other cells. Their shape and structure influence signal processing and determine connectivity patterns within neural circuits, while the scope of dendritic arbors defines the size of a neuron's receptive field. Dendritic morphology is also a key distinguishing characteristic of

**Fig. 7 | Connectivity of the AMd.core domain, domain networks, and single neuron validations. a** Multiple tracer injections display the AMd periphery/core organization. **b** Connections of AMd.core. **c** Labeled terminals and cells in MOs-fef following AAV (left) and CTB (right) AM injections (AMd↔MOs-fef). **d** PHAL/CTB MOs-fef co-injection confirms AMd↔MOs-fef reciprocity and reveals specificity of the connection with AMd.core. Note the periphery/core structure. **e** Cre-dependent AAV tracing of AMd.core neurons validates projections to MOs-fef (left) (AAVretro-Cre in ACA), with quantified data showing that the strongest projections to MOs stem from AMd.core, compared to AMd.ventromedial, AMd.medial tip/dorsomedial, and AMv (right). **f** PHAL AM injection labels terminals in ECT, PERI, ENTl layer V (L5). A retrograde tracer ENTl L5 injection (dashed circle at ARA 97) labels neurons in AMd.core, showing that projections to ENTl originate from AMd.core. **g** Cre-dependent AAV tracing of AMd.core neurons (AAVretro-Cre in ACA) shows consistent projections to ECT, PERI, and ENTl L5. **h** Cre-dependent AAV injection not tracing AMd.core neurons shows no labels in ECT, PERI, or ENTl. **i** Quantification of projections from neurons traced from AMd.core, AMv, and AMd.dorsal/dorsolateral reveals that the strongest projections to ENTl, PERI, and ECT arise from

AMd.core, with no projections to ENTm. **j** Hierarchical clustering of Cre-dependent AAV tracing from AMd domains to medial prefrontal and higher-order cortical areas. Output validates the unique or preferential connectivity of the proposed domains indicated by the asterisks. **k** Hierarchical clustering of projections from injections made in medial versus lateral AMd ($n = 1$ each) validate their distinct connections. Medial injection shows strongest projections to ILA and PL, while the lateral targets RSPv, RSPd, and PTLp. **l** Schematic of AM connections with ROIs that process visual/spatial information (top). Schematic of AMd.medial connections with MPF areas shown in Supplementary Fig. 4k. Dashed gray line represents the approximate division between ventral-medial domains that connect with limbic regions like MPF, HPF, amygdala (bottom left), and the dorsal-lateral AMd domains that connect with visual/spatial processing areas like ACA, RSP, PTLp (bottom right). **m** Normalized projections from axon reconstructions of traced neurons in AMd.medial, AMd.dorsal/dorsolateral, and AMd.core correlate with the mesoscale connections, showing that the AMd.medial neuron projects mostly to PL and ILA, the AMd.dorsal/dorsolateral to ACAv and RSPv, and the AMd.core to MOs. See Supplementary Table 2 for structure abbreviations.

neuronal types, although this information is limited for ATN. MORF3 genetic sparse labeling was combined with brain clearing and 3D microscopic imaging to obtain brain images through the ATN. These images showcased sparsely labeled excitatory ATN projection neurons with detailed dendritic morphology (Fig. 10a–d), which were then digitally reconstructed (Fig. 10e). This approach enabled systematic characterization of the morphological attributes of AD, AV, and AM neurons using computational methods.

Statistical comparisons of dendritic morphological features were carried out for a total of 109 neurons (AD = 10, AV = 17, AM = 82). The two-sided Wilcoxon signed rank test was used for pairwise comparisons, and false discovery rate (FDR) corrections were carried out for multiple comparisons. The results showed that AM dendritic arbors had greater dendritic length and a higher number of branches than both AD ($p < 0.001$ for both length and number of branches) and AV dendritic arbors ($p < 0.001$ for length and $p < 0.01$ for number for branches) (Supplementary Fig. 7a, b).

While AD and AV neurons did not differ much in length, AD neurons had greater branching asymmetry (i.e., disproportionate distribution of dendritic tips between the two daughter branches at branch points) than both AM and AV neurons (Supplementary Fig. 7c). While AD neurons had reduced average branch order (a measure of complexity, where the branch order of every compartment in a dendritic tree is averaged) compared to AV neurons ($p < 0.05$), AD neurons had increased height ($p < 0.05$) and increased maximum dendritic path length ($p < 0.05$) compared to AV neurons (Supplementary Fig. 7d–f). Due to this asymmetric dendritic extension, the AD neurons matched AM neurons in height and maximum path distance, even though AM neurons were much larger and more complex (Supplementary Fig. 7e, f). The average Sholl intersection profile also showed that the AV neurons had a higher number of intersections closer to soma compared to the AD neurons which extended out to a greater path distance away from the soma, matching the AM neurons in extension while being much lower in the number of intersections per concentric circle drawn at 50 μm intervals (Supplementary Fig. 7g). Comparative morphometrics suggest that AD neurons send out their dendritic arbor over specific directions in an asymmetric manner, whereas the AV neurons have a higher branch density closer to the cell body and sample the dendritic field more uniformly.

We also classified all neurons from ATN based on dendritic arbor size (total dendritic length) and complexity (total number of branches). All ATN neurons could be distinguished into two morphological types using K-means clustering. Type 1 neurons (84 out of 109) included relatively smaller neurons with lower numbers of branches (average dendritic length $3645 \pm 1428$ μm, average number of branches: $51 \pm 16$) (Supplementary Fig. 7 h). Type 2 (25 neurons) included larger and more branched neurons (average dendritic length

$8976 \pm 1998$ μm, average number of branches: $126 \pm 27$) (Supplementary Fig. 7h). Previous studies have identified bushy and radiate thalamic projection neurons based on prominent morphological features such as size and complexity[47–50]. Comparing previous neuron types with the current study suggests that radiate neurons with reduced numbers of branches may be analogous to the Type 1 neurons. The bushy neurons, with greater length and more branches are like the Type 2 morphology. All Type 2 neurons belonged to the AM region, whereas Type 1 neurons were from all three ATN nuclei. Finally, we also observed the grape-like appendages on the projection neurons that have been reported[48,51–53] (Supplementary Fig. 7i).

## Edge versus non-edge AMd neuron morphology

Neurons with somas lying predominantly on the edges of the AMd displayed a distinct morphological appearance (Fig. 10f, i). These edge neurons displayed relatively less dendritic wiring, and the dendrites seemed to have an orientation bias towards the center of the AMd (center-tropism). To quantify and test these observations, we assigned each neuron as edge or non-edge based on their soma location within the AMd. Comparisons of quantified morphological features between the two groups were made using a two-sided t-test with FDR corrections.

Edge neurons were generally smaller (reduced dendritic length, $p < 0.05$) than neurons with somas located towards the center (non-edge) of the AMd (Supplementary Fig. 7j). To evaluate and quantify the orientation bias (towards the AMd center) of the dendritic arbors, we calculated two features: (a) average angular deviation from the AMd center (Fig. 10j–l) and (b) proportion of dendrites closer to the AMd center than the soma (Fig. 10m). We measured each dendritic compartment's angular deviation relative to the local AMd center vector (i.e., the vector connecting the compartment's origin point to the AMd center, see "Methods" for details) (Fig. 10j). Averaging that value across all dendritic compartments produces the average angular deviation of the whole neuron. The average angular deviation provides an inverse estimate of an AMd neuron's bias toward the AMd center. That is, the smaller the average angular deviation, the greater the orientation bias towards the AMd center (Fig. 10k). Edge neurons had significantly lower average angular deviation compared to the non-edge neurons ($p < 0.0001$) (Fig. 10l).

We also measured the proportion of the neuron's total dendritic wiring that is closer to the AMd center than the neuron's soma. Similarly, as the branch orientation result suggested, the edge neurons had a higher proportion of dendritic wiring closer to the center than the soma compared to the non-edge neurons ($p < 0.0001$), demonstrating an AMd center bias (Fig. 10m). Overall, the average angular deviation of neurons is inversely correlated with the distance from the AMd center (Pearson's correlation coefficient $= -0.65$, $p < 0.0001$), i.e., neurons lying further away from the AMd center demonstrate greater tropism

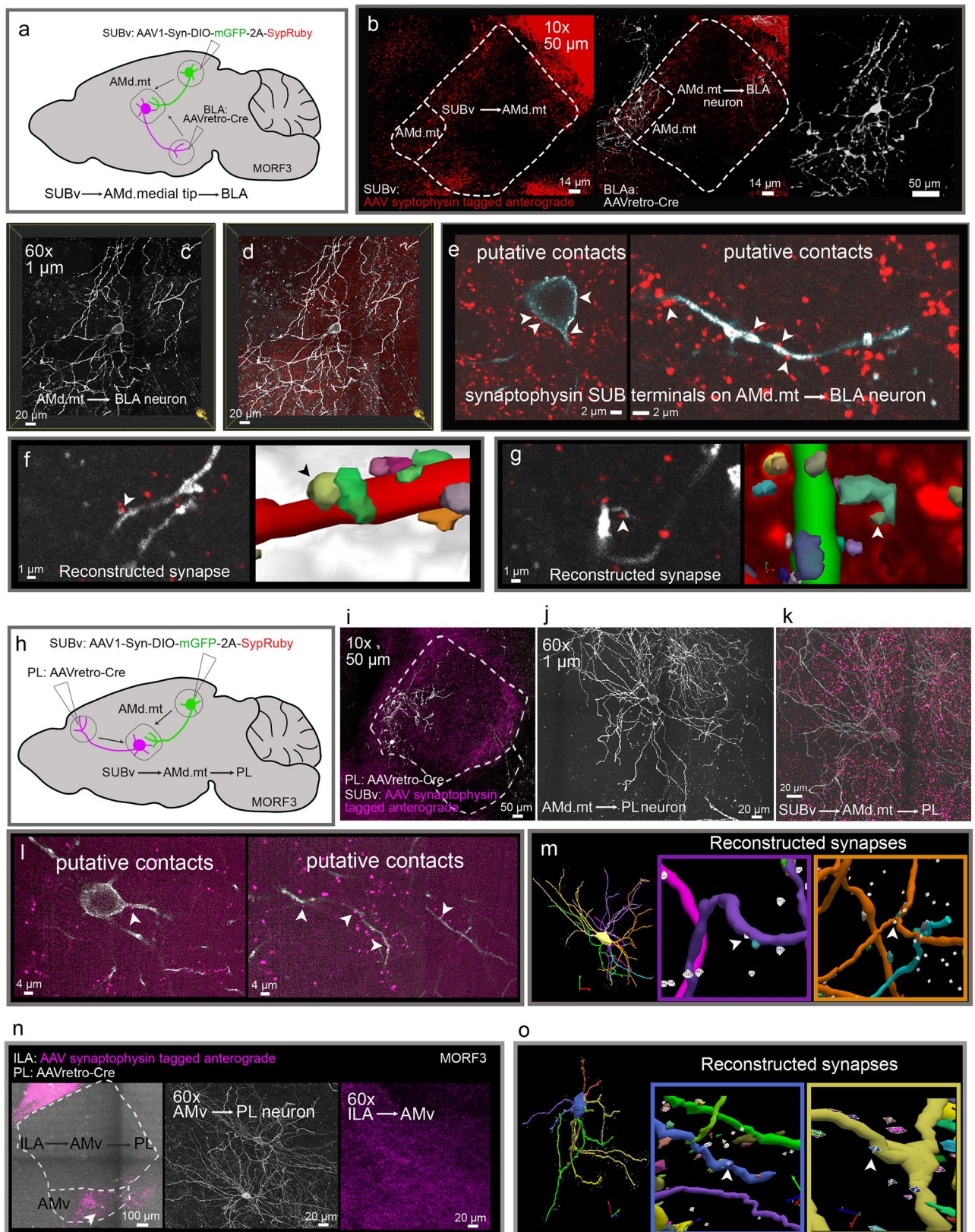

towards the AMd center (Fig. 10k). See "Methods" for details on all analyses conducted.

Like AM edge and non-edge neurons, AD and AV neurons whose somas were located at the edges of the nuclei qualitatively displayed a morphological appearance (i.e., stretched) that differed from their non-edge counterparts (Fig. 10g, h). These distinct features were not

quantified due to the low numbers of reconstructed neurons in each category.

**AMd domain level morphological comparison**
Neurons from AMd domains [AMd.core, AMd.dorsal (AMd.dorsal and AMd.dorsolateral), AMd.dorsomedial, AMd.lateral, and AMd.medial

**Fig. 8 | Medial prefrontal, hippocampal, and amygdalar synaptic circuits through AMd.medial tip and AMv. a** Injection strategy used in MORF3 mice to validate the SUBv→AMd.medial tip→BLA disynaptic circuit. **b** A synaptophysin tagged anterograde AAV injection made in the SUBv labels terminals in the AMd.medial tip, while an AAVretro-Cre injection in the BLA back labels neurons in the AMd.medial tip. **c** 60x magnification of a BLA projecting AMd.medial tip neuron. **d** The same AMd.medial tip→BLA neuron in (**c**) merged with labeled synaptic terminals from the SUBv. **e** Note the close apposition of the neuron processes and terminals suggesting putative synaptic contacts of SUBv terminals onto the BLA projecting AMd.medial tip neuron. **f, g** Putative contacts were reconstructed and validated via Neurolucida (see "Methods" for details). **h** Injection strategy used in MORF3 mice to validate the SUBv→AMd.medial tip→PL disynaptic circuit. **i** A synaptophysin-tagged anterograde AAV injection made in the SUBv labels terminals in the AMd.medial tip, while an AAVretro-Cre injection in the PL back labels neurons in the AMd.medial tip. **j** 60x magnification of a PL projecting AMd.medial tip neuron. **k** The same AMd.medial tip→PL neuron in (**j**) merged with labeled synaptic terminals from the SUBv. **l** Close apposition of neuron processes and terminals suggests putative synaptic contacts of SUBv terminals onto the PL projecting AMd.medial tip neuron. **m** Putative contacts were validated via Neurolucida same as in (**f–g**). **n** The same injection strategy in a and h was applied in MORF3 mice to validate the ILA→AMv→PL circuit. **o** Synaptic contacts of ILA projections onto PL projecting AMv neurons were reconstructed and validated. See Supplementary Table 2 for structure abbreviations.

(AMd.medial tip and AMd.ventromedial)] also demonstrated distinct levels of center bias, both in terms of average angular deviation and proportion of dendrites closer to core than soma (Supplementary Fig. 7k, l). Pairwise comparisons (two-sided Wilcoxon signed rank test with FDR corrections) showed that AMd.core neurons demonstrate a higher average angular deviation (lower AMd center-bias) relative to their local AMd center vectors compared to AMd.dorsomedial and AMd.medial domains ($p < 0.05$) (Supplementary Fig. 7k). Both the AMd.core and the AMd.dorsal neurons had a relatively lower proportion of dendrites closer to the center (lower AMd center-bias) compared to the AMd.medial neurons ($p < 0.05$; Supplementary Fig. 7l).

AMd domains also demonstrated distinct levels of average arbor height and maximum path length. AMd.dorsomedial neurons had reduced height, compared to core, lateral, and medial neurons. Medial neurons also showed greater height than dorsal neurons ($p < 0.05$; Supplementary Fig. 7m). Both medial and lateral neurons had greater maximum dendritic path distance (path distance of the dendritic terminal that is furthest away from the soma) than both dorsal and dorsomedial neurons ($p < 0.05$; Supplementary Fig. 7n). The fact that medial and lateral domain neurons that showed greater bias toward the center (i.e., showed lower angular deviation and higher proportion of dendrites closer to the soma) also tended to have greater height and maximum path length, corroborated the initial qualitative observation that neurons lying away from the AMd center tend to look stretched.

## Discussion
### Summary of findings and overview of the ATN functional neural network
The ATN are a key component of the classic Papez circuit (Fig. 11a), which has influenced our understanding of the neural basis of emotion and memory. Over time, the Papez circuit has evolved from a simple model of emotional processing to a more complex framework that integrates cognitive functions. Research has emphasized the critical roles of these structures in not only emotion, but also episodic memory, attention, and spatial processes (for review, see refs. 2,3). Accordingly, we present an updated and expanded version of the Papez circuit that subserves cognition and emotion (Fig. 11a, c). Specifically, we show that distinct ATN domains act as network hubs, bridging communications between (1) the cortico-basal ganglia system, which controls spatial orientation, navigation, and goal-directed behaviors[44], and (2) the classic limbic system (SUBv, amygdala, MPF), which regulates neuroendocrine, autonomic and social behavior associated with emotion[54,55]. These interactions are mediated through several major pathways, as discussed below, and depicted in a wiring diagram in Fig. 11c.

Both AD domains, the AD.medial and AD.lateral, receive direct inputs from the LM and DTN. While the AD.lateral shares reciprocal connections with the RSPd and RSPv, the AD.medial densely projects to the PRE, POST, and PAR, which, in turn, project back to the LM to complete a circuit loop with the AD and DTN. Each structure within this network houses head-direction cells[17], pivotal for spatial orientation.

The AV subdivisions (AV.lateral, AV.dorsal, AV.medial, and AV.medial tip) are connected with the MM and VTN, forming a central network that regulates theta rhythmicity to synchronize activity across structures to facilitate communication relevant for learning and memory[22,56]. The AV.dorsal shares reciprocal connections with the subicular complex (PRE, PAR, POST) and with RSPd and RSPagl. The AV.lateral receives input from the SUBd, but generates projections to the SUBv, which in turn projects back to the AV.medial and AV.medial tip. These latter AV domains share reciprocal connections with the RSPv, a critical component of the medial cortical network[44] that also receives direct inputs from the SUBd. The AV also receives input from the LDT nucleus, implicated in REM sleep, attention, and reward processes.

We identified five distinct subdomains within the AMd, each also interconnected with parts of the medial cortical network[44], including the RSPv, PTLp, ACAv, ACAd, and its adjacent MOs-fef. Thus, information processed regarding head direction, theta rhythm, attention, and REM sleep from the ATN converges onto the medial cortical network, within which this information is integrated with other external environmental cues, including visual, auditory, and somatosensory inputs, and higher-order associative cortical information[44]. These medial neural network cortical areas project densely to the SC, zona incerta (ZI), and CPdm, which projects to the ventromedial division of the substantia nigra pars reticulata (SNr), and the SNr in turn projects to the medial SC[37,57]. Together, these regions establish a core neural network regulating eye, head, and neck movements[36] that are important for attention, spatial orientation, navigation, and exploratory behavior.

Further, the AMd.medial and AMv are closely linked with the MPF (PL, ILA), SUBv, and BLAa—three limbic cortical areas that regulate social behavior and emotion-related activities through their projections to: (1) The medial amygdalar nucleus (MEA) and the posterior part of the bed nuclei of the stria terminalis (BSTp), which generate dense projections to the hypothalamic medial behavioral control column composed of the anterior hypothalamic (AHN), ventromedial hypothalamic (VMH), and dorsal premammillary nuclei (PMd)[58]. This hypothalamic subnetwork projects extensively to the dorsal PAG (PAGd), governing goal-directed behaviors with strong emotional components, such as hunting and attacking[58], which require attention and navigation during their execution. Notably, the PMd, which functions as a threat detector by sensing dynamic changes under threatening conditions as the animal approaches and avoids the threatening source[59], projects densely back to the AMv, which is crucial for updating memory processes to adapt to changes under threatening conditions. (2) The central amygdalar nucleus (CEA) and anterior BST (BSTa), which are involved in the regulation of autonomic and neuroendocrine activities[58]. (3) The nucleus accumbens (ACB), which, along with the ventral tegmental area (VTA), controls reward mechanisms and addiction.

### AMd.medial and AMv as communication subnetworks for MPF, HPF, and amygdala
A network of structures, including the cortex, hippocampus, and thalamus are critical for the formation of memories. The ATN play a

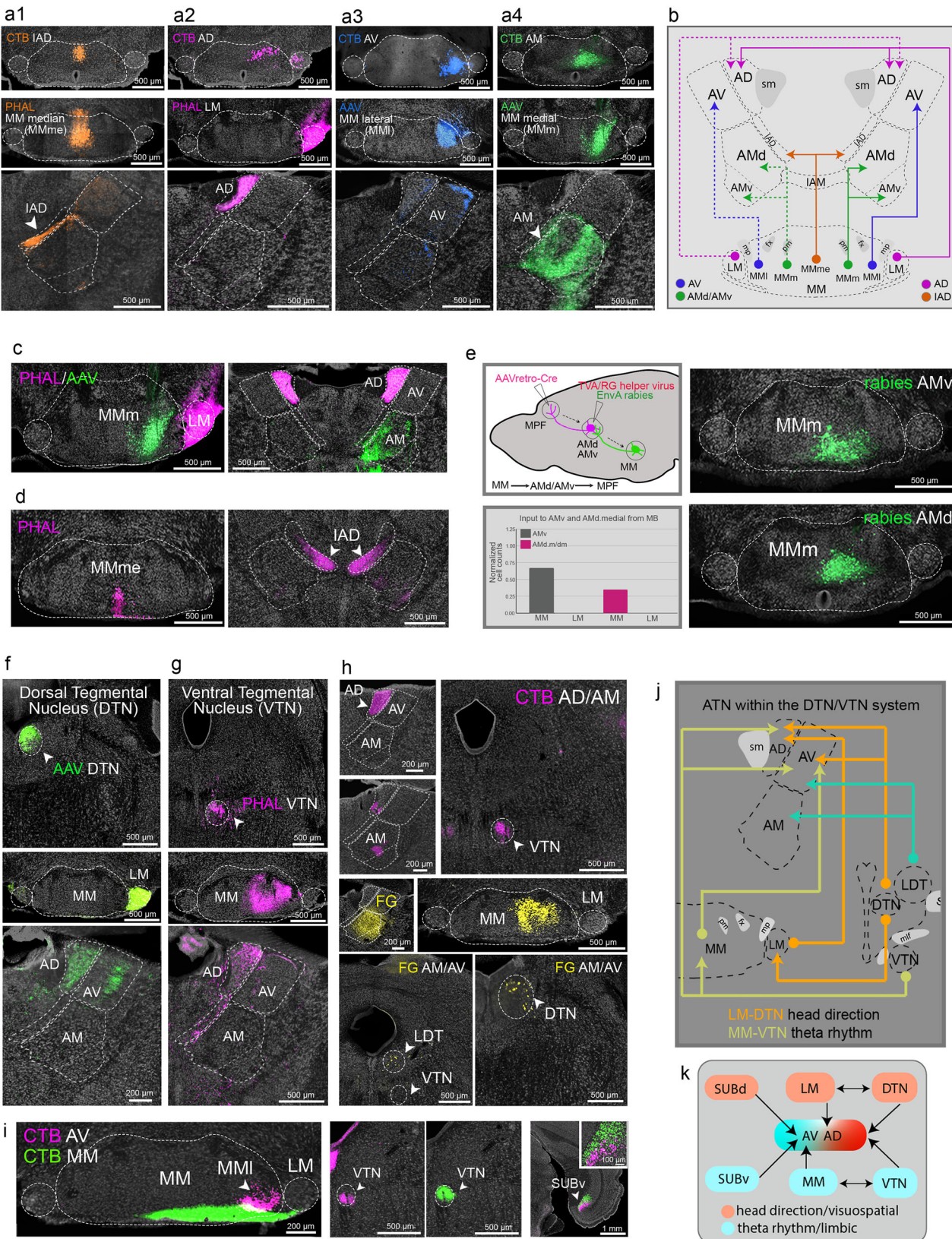

critical role in this network, but this is especially true for the AM since it is the only one connected with both the hippocampus and MPF. The AMd also connects with the BLA, shown here to be specifically the anterior BLA (BLAa; Fig. 6f). Importantly, cortical, hippocampal, and amygdalar connections are relevant for the formation of emotional memories including fear-related behavior[60], and AM lesions disrupt

contextual fear responses[29,30]. We found a specific subnetwork within the AMd through which these cortico-hippocampal-amygdalar structures communicate: SUBv→AMd.medial→PL/ILA and SUBv→AMd.medial→BLAa (Fig. 8a, h). Completing this network are the strong PL/ILA↔BLAa reciprocal connections[38]. These clearly show AMd.medial as a hub for cortico-hippocampal-amygdalar interactions.

**Fig. 9 | ATN connections with hypothalamus, midbrain, and hindbrain.**
**a** Projections from mammillary bodies to ATN. **a1** A retrograde CTB injection in IAD labels neurons in MM median (MMme; MMme→IAD) (top). A PHAL anterograde tracer injection in the MMme (middle) labels the IAD (bottom) validating the connection. **a2** CTB injection in AD labels LM neurons (MMl neurons labeled from injection leakage into AV). The LM→AD is validated with a PHAL LM injection that labels the AD. **a3** A CTB AV injection labels MMl neurons. This MMl→AV connection is validated by an MMl AAV injection. **a4** A CTB AM injection labels MMm neurons (MMm→AM), which is validated by an AAV MMm injection. **b** Schematic summarizing ATN, MM, and LM connectivity. **c** PHAL and AAV tracers in MMm and LM in the same brain show the bilateral LM→AD projections compared to the unilateral MMm→AM projection. **d** MMme PHAL injection shows bilateral MMme→IAD projections. **e** Cre-dependent TVA receptor mediated rabies tracing of AMv and AMd.medial neurons retrogradely label MMm validating MMm→AMd/AMv connections. Bar graph shows the quantification of retrogradely labeled cells in the MM

versus LM following the injections. **f** AAV injection in the DTN (top), confirmed by strong projections to LM (middle), labels terminals in AD and AV (bottom). **g** AAV in the VTN (top), confirmed by the strong projections to MM (middle), labels terminals in AD and AV (bottom). **h** Validation of the DTN/VTN→AD/AV connections. CTB injection in AM/AD labels cells in VTN, while a FG AM/AV injection, confirmed by MM labeled cells, shows labels in DTN and LDT. **i** Double retrograde injections in AV (CTB, pink) and MM (CTB, green) show labeled cells in VTN (VTN→AV/MM) and in SUBv (SUBv→AV/MM). Note the labeled cells in MMl, which confirm AV injection location. Also, note the laminar specific arrangement of SUBv→AV and SUBv→MM projecting neurons. **j** Schematic of connections among ATN, DTN, and VTN. **k** Schematic showing generally segregated networks of head directionality/visual-spatial processing (peach) involving AD←LM↔DTN and theta rhythmicity (blue) involving AV←MM↔VTN. Direct connections shown in (**f**, **g**) suggest a more interconnected network rather than segregated circuits. See Supplementary Table 2 for structure abbreviations.

In addition, we show that the AMv is strongly connected with MPF areas like the rostral ACA, PL, ILA, and ORB (Fig. 6k) and the ILA→AMv→PL circuit was specifically shown at the synaptic level (Fig. 8n). These AMv connections are also likely relevant in the formation of emotional memories given the demonstrated role of corticotropin-releasing factor containing AMv neurons in the regulation of fear conditioning[31].

Notably, AM–MPF connections are linked to goal-directed behaviors involving intracranial self-stimulation[27]. Specifically, stimulation of MPF terminals in the AM activates midbrain dopaminergic neurons in the ventral tegmental area (VTA) and reinforces intracranial self-stimulation. A similar outcome results from stimulation of AM terminals in the MPF. Inhibition of these VTA neurons reduces the self-stimulation facilitated by the MPF-AM connections. This finding is interesting in the context of a wider AM network, specifically for the AMd.medial subnetworks. The projections from the AMd.medial tip to the BLAa are to its caudal domain (BLA.ac) (Fig. 6f), which is shown to be in a network of structures involved in drug seeking behavior[38] (Fig. 11b). These structures include the SUBv, ILA/PL, medial accumbens (ACB), medial olfactory tubercle (OT), rostral paraventricular thalamic nucleus (PVT), and CA3. Figure 11c also demonstrates the wider network of the MPF and AMd.medial that lead to the VTA.

## Direct connections between the ATN, DTN, and VTN

Two parallel circuits DTN↔LM→AD and the VTN↔MM→AM/AV are respectively involved in the head direction system and in theta rhythm activity. A surprising finding was direct connections between the midbrain DTN/VTN nuclei and the AD/AV (Fig. 8f, g). Given the reported highly segregated tegmental-mammillary pathways, this result was unexpected and suggests some level of convergence across the two processing streams. Within the context of confluence, these direct connections are less surprising since the idea of head directionality and theta rhythmicity coalescing with the ATN is also suggested by the presence of head direction cells in the AV[16,18]. Interactions between head directionality and theta rhythmicity also seem intuitive: the synchronization of activity across regions involved in head directionality would facilitate an animal's ability to learn and retain spatially pertinent information essential for navigation. While no direct connections between ATN and VTN/DTN have been documented, there is a study in humans that potentially lends support for this[61].

## ATN connectivity reported in our findings and the current body of literature

Studies across different species have identified mammillary body and hippocampal connections primarily exclusive to AM, AD, and AV that form parallel, segregated loops across the Papez circuit subserving distinct functions in navigation[1,9]. Projections from distinct regions of the medial (MM) and lateral (LM) mammillary nucleus to each of the ATN nuclei are well documented in rats and non-human primates, with

the MMm projecting mostly to AM[62,63], the MMl to AV[64], and the LM to AD[63,65]. Projections from the MM medianus (MMme) to IAM are also reported[15]. Our data show these precise MB→ATN connections, although we also show MMme→IAD projections (Fig. 9a–e).

Aside from the mammillary body projections to ATN, many inconsistencies are reported regarding ATN connectivity, likely due to different methods, species, injection sizes, injection site locations, and different anatomic nomenclature that have been utilized across the studies, underscoring the importance of systematically collected data. Hippocampal connections with individual ATN nuclei are similarly segregated with many conflicting connections reported. Some show AD projecting to postsubiculum (POST) and AV and AM projecting to proximal (near CA1) and distal subiculum, respectively[66,67], while others show PRE/SUB→AM/AV and POST/PAR↔AV[15]. POST/PRE/PAR↔AV/AD, AM→PRE, and AD↔hippocampus connections also have been reported[68]. Our collective data of ~200 traced pathways showed specifically (1) AD.medial→POST/PRE/PAR with no projections back to AD (Fig. 2b, e, f); (2) AV.dorsal↔POST/PRE/PAR (Figs. 2f and 3b–d); and (3) no connections between the AM and POST, PRE, and PAR (Fig. 1f). Regarding the SUB, we showed (1) no connections between SUB and AD (Fig. 2g); (2) SUBv→AV.medial tip (Fig. 3k); (3) SUB-d→AV.lateral→SUBv (Fig. 3j, k); (4) SUBv→AMd.medial tip (Fig. 5b, c); and (5) SUBv caudal↔AMd.lateral (Figs. 3l and 5l). No connection between the ATN and the CA1, CA2, nor CA3 were observed.

Different combinations of connections have been reported between the ATN and ENT like ENT→AM[42], AM→ENT[33,34], and AD/AV/AM→ENT[28]. We only found an AM→ENT connection. Many anterograde and retrograde injections were placed across the ENTl and ENTm (Supplementary Table 1). The large majority of ENTl tracer injections did not produce any labeling in the AM. This was because the AMd projects to a specific region within ENTl layer V (Fig. 7f, g). Once a retrograde tracer injection was successfully placed in that specific ENTl region, some AMd.core neurons were labeled (Fig. 7f). Our AM injections did not produce any labels in the ENTm despite reports of these connections in the literature[33].

ATN connections with cortical regions are reported to be separable with the ventral (granular) RSP (RSPv or RSPg) connecting mostly with AD and AV, the dorsal or agranular RSP (RSPd or RSPagl) with the AM[67,69,70], and the PL and orbital (ORB) with the AM[28,34]. Our data showed (1) RSPd/v↔AD.lateral (Fig. 2c, d); (2) RSPd/agl↔AV.dorsal (Fig. 3g; Supplementary Fig. 2e) (3) RSPv↔AV.medial (Fig. 3e, g); and (4) RSPv←AMd.dorsolateral (Fig. 5a). Sparse RSP-AM connections were detected, and ACA connections were exclusively with the AM despite reported ACA connections with AV and AD (Fig. 4k)[28,42,71]. Specifically, AMd.dorsomedial/AMv↔rostral ACA and AMd.dorsal↔caudal ACA (Fig. 4k; Supplementary Fig. 3b) connections were found. The exclusive AM-ACA connections reinforce the AM's role in emotional learning given that the ACA is involved in emotional processing, fear acquisition memory, and in the control of innate fear responses[72–74]. Further, a

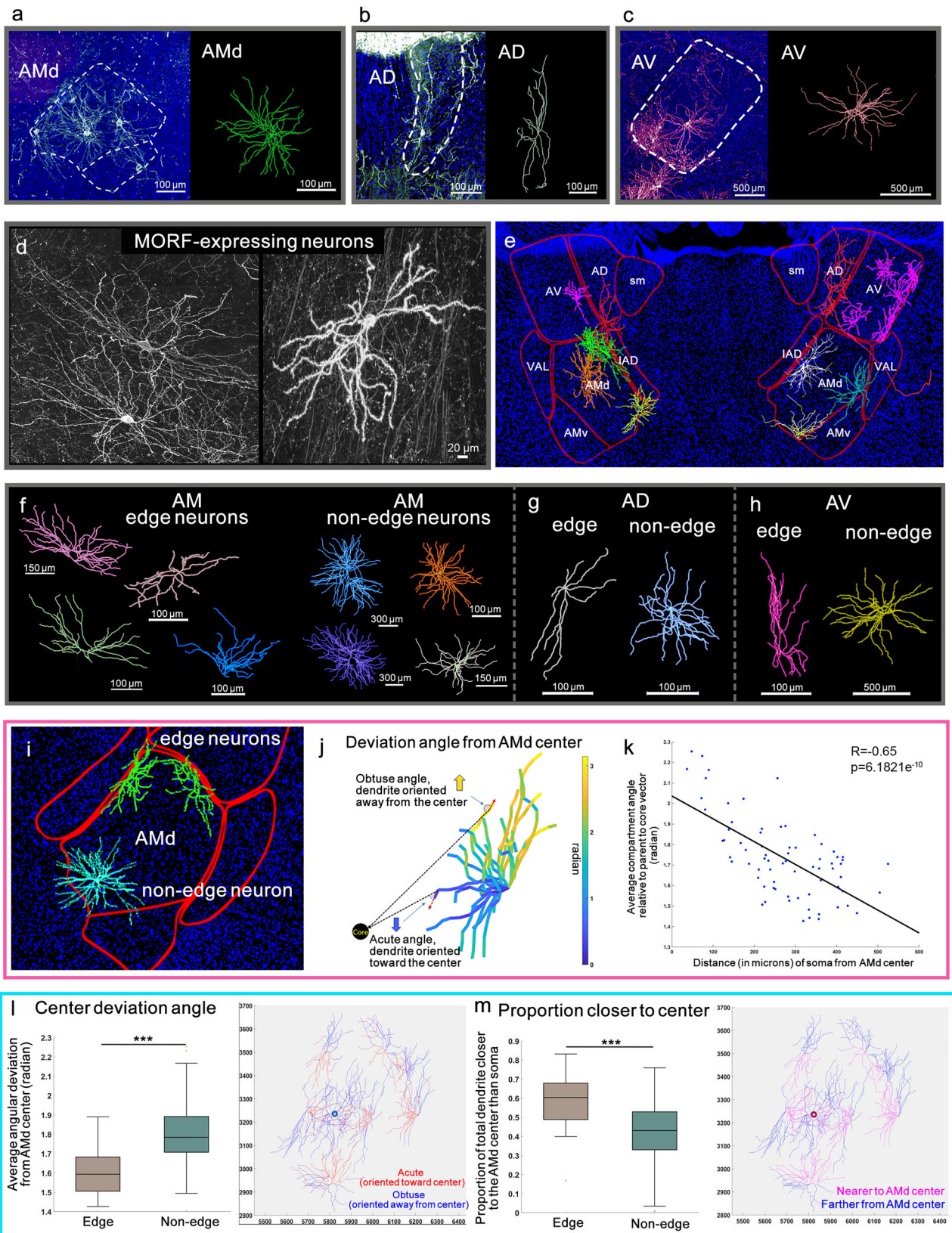

rostral versus caudal distinction of the ACA is generally accepted, with distinct respective roles in emotional versus cognitive processing[75–77].

Our data generally agree with the literature regarding connections of the PL, ILA and ORB being exclusive to AM; however, our data greatly expand upon this by demonstrating AM domain level connections with these medial prefrontal cortical areas, and also the dissociable connections of the AMd and AMv, which have not been extensively reported[31]. We show (1) AMd.medial tip/AMv⇔PL; (2) AMd.ventromedial/AMv⇔ILA; and (3) AMd.ventromedial/AMv⇔ORBm/vl (Fig. 6b–d, g–h). We did not find significant AMv projections to BLA, RSP, ECT, PERI, ENTl, nor to CA1, which have been reported for AMv CRH neurons[31], and to our knowledge

**Fig. 10 | Dendritic morphology analysis of ATN neurons.** Representative images of (**a**) AM, (**b**) AD, and (**c**) AV neurons and their corresponding representative digital reconstructions. **d** MORF expressing ATN neurons. **e** Representative dendritic arbors from the ATN, where edge (left panels) and non-edge (right panels) neurons are displayed for (**f**) AM, (**g**) AD, and (**h**) AV. **i** Location and orientation of AMd edge (green) and non-edge neurons (teal). **j** Graphical definition of a compartment's angular orientation relative to the local AMd center vector: dashed straight lines that connect a compartment's origin point to AMd center. 1 radian = 57.3°. **k** Compartment deviation angle from the AMd center as a function of the somatic distance from AMd center. The greater the distance of a neuron's soma from the AMd center, the lesser its average compartment deviation ($R = -0.63$, $p = 6.1821e^{-10}$). **l** Edge neurons ($n = 27$) show reduced angular deviation from the AMd center compared to non-edge neurons ($n = 45$) (two-sided $t$-test with FDR correction; $t(70) = -5.4403$, $p = 7.378e^{-07}$). Edge neurons have a higher proportion of dendritic compartments oriented toward AMd center. Red colored dendrites have an acute angular deviation and are oriented towards AMd center as opposed to the blue dendrites that have an obtuse angular deviation and are oriented away from AMd center (right panel). The $x$-axis increases from medial to lateral, while $y$-axis increases from dorsal to ventral (in microns). **m** Edge neurons have a higher proportion of total dendrites closer to the center than their individual cell bodies [$t(70) = 4.353$, $p = 4.4886e^{-05}$]. Pink designates closer dendrites, while blue dendrites are further away (right panel). Traced AM, AD, and AV neurons were collected across independent cases and aggregated for analysis. The line inside of each boxplot is the sample median. Top and bottom edges are upper and lower quartiles, respectively. The distance between top and bottom edges is the interquartile range (IQR). Upper quartile corresponds to the 0.75 quantile and lower quartile corresponds to the 0.25 quantile. Outliers ("o") are values that are more than 1.5 IQR away from the top or bottom of the box. Top whisker connects the upper quartile to the nonoutlier maximum, and the other connects the lower quartile to the nonoutlier minimum. ***$p \leq 0.001$.

a study dissociating the brain-wide inputs to the AMv has not been conducted.

Regarding ATN connections with secondary motor cortical areas (MOs), our data showed selective reciprocal connections with the AMd.core (Fig. 7b–e), and although these connections have been reported for the AM[33], AV↔MOs connections also are reported[28]. The MOs is a large, undivided structure in most rodent atlases, and it is possible this discrepancy is due to differences in MOs injection site locations. We placed injections across the entire MOs, and only the injections specifically in the MOs-fef region (adjacent to the ACAd)[39,44] produced labeling in the AM. Notably, this MOs-fef region projects to the CP dorsomedial region, which integrates information from a variety of visual areas including VISam, VISal, ACAv, and PTLp caudal medial[39] (Supplementary Fig. 6f). We also show AM→caudal ECT/PERI connections (Fig. 7f, g), both of which heavily connect with cortical areas involved in visual and auditory processing[44].

Finally, despite documented connections between the internal part of the globus pallidus with the AD[78], these connections were not found through our dataset.

## Caveats to tracing experiments

Our large dataset used for the current project utilized a variety of chemical and viral tracers. Importantly, each tracer has distinct characteristics and exhibits varied neurotropism that can meaningfully affect connectivity results. For example, anterograde AAVs label fibers of passage, while PHAL does not. AAVretro preferentially labels the cortex, while rabies preferentially labels hypothalamic neurons[79]. These differences underscore the importance of data validation. In this work, the connectivity data were validated in multiple ways to ensure the reliability of the results. Retrograde tracers were placed in regions of anterograde terminations, while anterograde tracers were placed in regions of retrogradely labeled cells. For example, anterograde tracer injections in the AM and AV show labeled terminals in the SUBv, but primarily in the caudal regions (Supplementary Fig. 4a). Retrograde tracers in the rostral SUBd, rostral SUBv, and the caudal SUBv confirm that only the caudal SUBv projects to the AM and AV (Supplementary Fig. 4b). Our data was further validated with Cre-dependent tracing methods and with repeated injections made in each ROI (e.g., Supplementary Figs. 3b and 4c). Together, these validation studies increase the confidence in the reported connections and mitigate potential issues arising from individual tracer characteristics. Notably, we have previously shown that repeated injections even in the smallest of ROIs produce similar brain-wide labeling patterns regardless of the tracer used[38].

On a final note, the accurate assessment of injection site location is also a critical component of connectomics. Generally, the exposure times used to image and capture labeled fibers and cells typically oversaturate the injection sites. As such, we reimage injection sites with lower exposure parameters to gauge its location and spread more accurately. The reimaging, along with all the data validation experiments, facilitate the accurate assessment of injection site location. For example, an anterograde AAV AM tracer injection labeled fibers and terminals in the CP and the ACA (Supplementary Fig. 6a, b). Retrograde tracer injections in the CP were placed precisely in the region of the AAV-labeled fibers, back-labeled neurons in the adjacent IAD and not AM (Supplementary Fig. 6a, b) suggesting an IAD→CP connection and not an AM→CP connection. Retrograde ACA tracer injections on the other hand showed labeled neurons in the AM (Fig. 4e, f).

## ATN morphology

Identification of the morphological neuron types based on their prominent dendritic features is necessary to understand the input-output relations within the ATN. Across the thalamus, the somatodendritic morphology of projection neurons and interneurons have been identified. Generally, four thalamic projection neurons (TPNs) or Golgi Type I neurons, whose axons extend outside their native thalamic nucleus to innervate the cortex, striatum, and amygdala, have been reported. The first are the bushy tufted TPNs, which are large with a high number of branches whose dendrites profusely arborize, are intertwined, and are covered in spines, which give them their bush-like appearance[50]. The second are the radial (or stellate) TPNs, whose dendrites arborize in a radial fashion and compared to the tufted TPNs display relatively reduced wiring with shorter branches, particularly in the distal dendrites[48]. These radiate cells often also display grape-like appendages close to the initial branch points of the primary dendrites[47,48,80], although these appendages are also reported for thalamic interneurons[51,80]. The tufted and radiate TPNs are found throughout the thalamus, while the third category of TPNs, the diffuse or reticulated, have been reported primarily for the parafascicular (PF)[47,81–84], ethmoid-limitans[47], and paralaminar group (medial division of the medial geniculate (MG) nucleus, posterior intralaminar, suprageniculate, peripeduncular)[85]. Diffuse TPNs are characterized by few, poorly ramifying primary dendrites that spread across long distances and have many spines. Although ATN somatodendritic morphology has not been systematically examined in any species, some investigations have been made in the AM and AV of the cat[86], camel[51], and human[52] through Golgi staining, and have identified both bushy and radiated TPNs and interneurons. Our $k$-mean analysis on the total branch number and total dendritic length identified two clusters of neuron types in the ATN: one larger with more complex branching and a second that are smaller and less complex. These two clusters potentially correlate to the bushy and radiate cells identified in other thalamic nuclei (Supplementary Fig. 7h). Visually, no reticulated cells were identified in the ATN, and grape-like appendages were observed in some instances (Supplementary Fig. 7i).

In addition to TPNs, the thalamus contains interneurons, or Golgi Type II neurons, whose axons do not extend out of the parent thalamic nucleus and make local connections. These interneurons are generally

## a. Classic Papez circuit

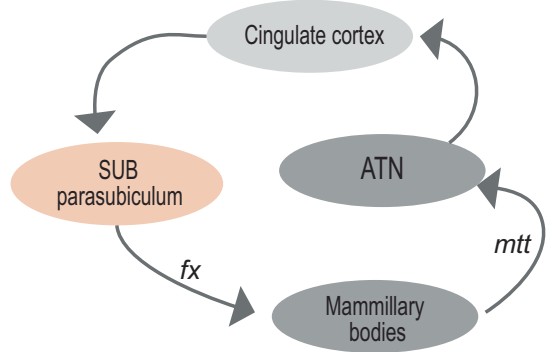

## b. Network of structures in drug seeking behavior

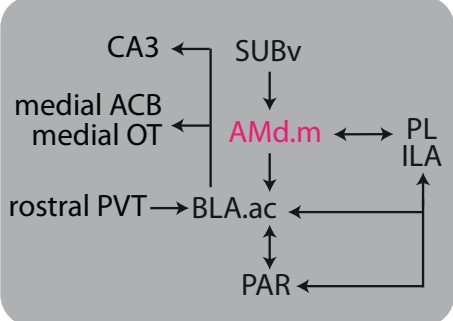

## c. Functional neural networks of the ATN

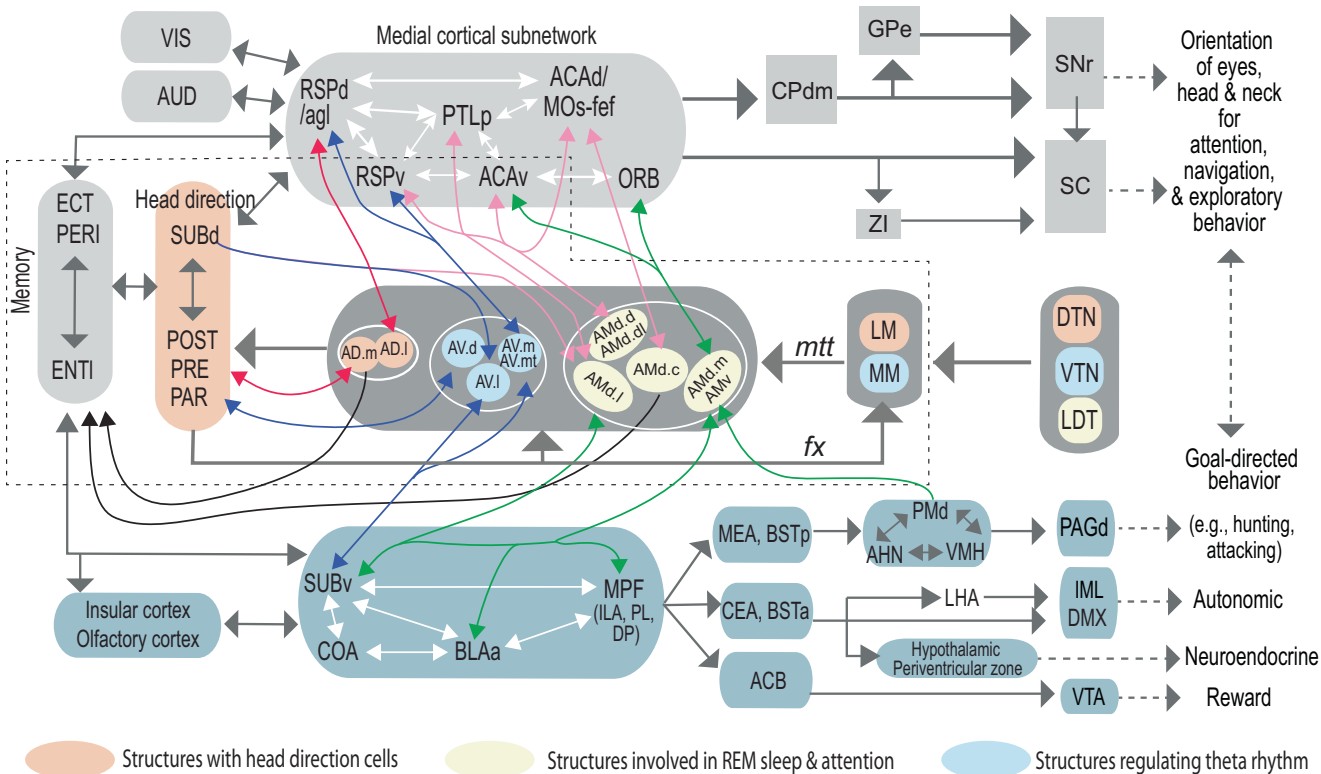

**Fig. 11 | Functional ATN networks. a** Depiction of the classic Papez circuit. **b** Network of structures involved in drug seeking behavior centered around the AMd.medial domains. Projections from the AM→BLAa are to a specific domain, BLA.ac, which is also connected with regions shown to be involved in drug seeking behavior (see "Discussion"). **c** Functional neural network of the ATN. An extended ATN wiring diagram showing how each of the identified ATN domains connects with different areas within the medial cortical subnetwork[44], which controls movements and the orientation of the eyes, head, and neck for attention, navigation, and exploratory behavior via the cortico-basal ganglia (SNr serves as the output portal) or through its cortico-tectal projections (SC serves as the output portal). Meanwhile, several ATN subnuclei (especially AMv, AMd.m, AMd.l, AV.l, AV.m, and AV.mt) connect with the MPF (ILA, PL, DP), BLAa, and SUBv. These three

cortical areas are highly interconnected and generate massive descending projections to the medial amygdalar nucleus (MEA), central amygdalar nucleus (CEA), anterior (BSTa) and posterior (BSTp) bed nuclei of the stria terminalis, hypothalamus, and PAG to regulate neuroendocrine, autonomic, and goal-directed behavior associated with emotion (e.g., hunting or attacking). Note that the dorsal premammillary nucleus (PMd) generates dense projections back to the AMv. Finally, these three cortical areas also generate dense projections to the ACB, which, together with the VTA, are essential for reward and addiction. The structures within the dashed box are included in the classic Papez circuit. As illustrated here, the ATN serves as a critical network hub, bridging communication between the medial cortical subnetwork and the emotional network to control goal-directed behavior. See Supplementary Table 2 for structure abbreviations.

smaller and have fewer primary dendrites that poorly ramify but are covered in spines. Due to our method, we did not label any ATN interneurons (e.g., AAVretro-Cre injections in MORF3 mice to label TPNs).

## Morphology and function

Dendritic topological differences between bushy and radiate thalamic neurons are likely indicative of functional variability. For example, two types of neurons in the motor thalamic VA and VL nuclei have been

identified. Neurons that receive predominantly inhibitory inputs display significantly fewer dendrites and their axons project to layer I of the cortex and to the striatum[87]. The neurons that receive predominantly excitatory inputs have more dense dendritic trees and their axons project solely to layers II-V of the cortex. Presumably, the former neurons correspond to radiate TPNs, while the latter to tufted ones[88,89]. The tufted and radiate cells of the MG have been similarly functionally differentiated[90].

Larger dendritic arbors allow for a greater number of presynaptic partners for a neuron. We observed that dendritic arbors from the AM region were larger and more complex compared to those in the AV and AD. We also observed that based on our morphological classification (using arbor length and number of branches as the two determining classifiers), all the Type 2 neurons (large and complex, analogous to bushy neurons) belonged to the AM regions. The greater length of the AM dendritic arbors would in principle allow them to form many more synapses than the AV and AD neurons. This suggests a greater level of synaptic integration (particularly from excitatory long range axons) occurring in AM compared to AV or AD. Similarly, for the AMd regions, the non-edge neurons had larger dendritic arbors than edge neurons, indicating a greater level of axonal input for the non-edge neurons. Apart from topological properties (such as length and number of branches), the spatial orientation of neuronal arbors have been shown to be relevant to their functional roles[91,92] and AMd edge neurons demonstrate an orientation bias towards the AMd center that increases as the distance from the center also increases. The higher overall length of dendritic arbors at the AMd center region, combined with the preferential orientation of the edge neuron's dendritic arbors towards the center suggests a functional role, where individual neurons are attempting to maximize their presynaptic input from the central parts of the AMd region. It also suggests an attempt to make well-defined, separate channels of information processing with minor spillover across border regions. The ATN anterograde/retrograde tracing data tend to show a high degree of topographic specificity, in many instances precisely innervating right up to the border of the target nucleus with minimal crossover (Fig. 4a). Simultaneously, the dendrites of the edge neurons likewise conform to this specificity, minimizing their spread into adjacent nuclei (Fig. 10e). The result is that each subnucleus has a particular combination of information it preferentially integrates, and the dendritic morphology of the neurons helps to maintain that segregation of information processing. To our knowledge, this type of morphological organization has not been reported.

## Methods
### Subjects
One hundred fifty 8-week-old C57Bl/6J (Jackson Laboratories) male mice were used to trace ~200 pathways (Supplementary Table 1). Animals were housed in pairs in a room with controlled temperature (21–22 °C), humidity (51%), and lighting (12-h light:12-h dark). Prior to stereotaxic surgeries for tracer delivery, mice were given at least 1 week to acclimate to their environment. Throughout the experiments, subjects had unrestricted access to tap water and mouse chow.

Seventeen male MORF3 mice from our established breeding colony at UCLA were used (1) to sparsely label neurons in AD, AV, and AM (including AM domains) for morphometric analysis and (2) to reveal the disynaptic circuitry through the AMd.medial tip (SUB-v→AMd.medial tip→PL/BLAa) and AMv (PL→AMv→ILA). MORF3 mice (C57BL/6-Gt(ROSA)26Sortm3(CAG-sfGFP*)Xwy/J), developed by Dr. X. William Yang's lab at UCLA[46], are a Cre reporter mouse line engineered to utilize a mononucleotide repeat frameshift (MORF) for in vivo cell labeling. These mice express a Cre-dependent tandem spaghetti monster fluorescent protein with 20 V5 epitopes (smFP-V5), preceded by a polycytosine repeat (C22) MORF switch, all under the control of a CAG promoter. Through Cre recombination and a spontaneous

frameshift mechanism, MORF3 mice enable sparse and stochastic labeling of neural cells, which can be visualized using V5 antibody staining.

Only male mice were used in this study. To our knowledge, sexually dimorphic connections are challenging to identify at the mesoscale resolution, predominantly employed here, particularly in the thalamus compared to more sexually dimorphic brain regions like the hypothalamus. Although we previously examined sex differences in mesoscale connections in certain brain areas, such as the basolateral amygdala, we did not observe any notable differences. Importantly, we aim to consolidate all our connectome research to establish the most comprehensive and reliable brain-wide mammalian connectome. Given that most of our connectivity experiments are conducted using male mice, maintaining consistency in our approach was essential.

### Ethics statement
We are committed to promoting ethical research practices and ensuring the welfare of all animals involved in our studies. All procedures adhered to regulatory standards as delineated in the National Institutes of Health Guide for the Care and Use of Laboratory Animals, as well as institutional guidelines set forth by the Institutional Animal Care and Use Committees at the University of Southern California (USC) and the University of California, Los Angeles (UCLA).

### Neural tracer injections
Anterograde and retrograde tracers were administered to anatomically defined regions throughout the brain to investigate their connectivity patterns. Stereotaxic surgeries for tracer infusions were conducted under isoflurane anesthesia. Initially, mice were anesthetized in an induction chamber containing isoflurane and then secured to the stereotaxic apparatus, where they remained under anesthesia via a vaporizer. Isoflurane was vaporized and mixed with oxygen (0.5 L/min), maintaining the percentage of isoflurane in the gas mixture between 2 and 2.5. Buprenorphine SR (1 mg/kg) was administered at the beginning of the surgery as an analgesic and ophthalmic ointment was applied to the eyes for protection from light. Tracers were delivered either iontophoretically (10 min 5 μAmp, 7-second alternating current) or via pressure injection (20–80 nl) using glass micropipettes with outside tip diameters measuring ~10–30 μm.

### Tracers
The tracers used to determine afferent and efferent connections included *phaseolus vulgaris* leucoagglutinin (PHAL, 2.5%; Iontophoresis, 10 min; Vector Laboratories); AAV-GFP (AAV1-hSyn-EGFP-WPRE-bGH; $2.7 \times 10^{13}$; Iontophoresis, 3 min; Addgene); AAV-RFP (AAV1-CAG-tdTomato-WPRE-SV40; $2.0 \times 10^{13}$; Iontophoresis, 3 min Addgene); Glycoprotein-deleted rabies (RVΔG) (Gdel-RV-4tdTomato and Gdel-RV-4eGFP; $9.6 \times 10^{10}$; Pressure, 50 nl; Ian Wickersham laboratory at MIT); Fluorogold (FG, 1%; Iontophoresis, 3 min; Fluorochrome); Cholera toxin subunit B-Alexa Fluor 488, 555, 647 conjugates (CTB, 0.1–0.2%; Pressure, 50 nl; Invitrogen); AAVretro-Cre (AAVretro-EF1a-Cre; $2.4 \times 10^{13}$; Iontophoresis, 5 min; Salk Institute). Typically, in a single brain, 2–4 of these tracers were injected in different combinations. For example, in the triple anterograde tracing method, three anterograde tracers were injected in three different brain regions to reveal topographic projections. In the quadruple retrograde design, four retrograde tracers are injected in four different brain regions. These multiple injections per brain assisted in revealing, validating, and clearly visualizing the ATN domains (e.g., Figs. 4h & 5b). In the double co-injection design, two co-injections of an anterograde/retrograde tracer cocktail are injected into two different regions to assess input/output of different regions and to assess the interaction of the two regions injected.

For Cre-dependent anterograde tracing, the following tracers were used: AAVretro-Cre (AAVretro-EF1a-Cre; $2.4 \times 10^{13}$; Iontophoresis,

5 min; Salk Institute); Cre-dependent AAV-FLEX-GFP (AAV1-CAG-Flex-eGFP-WPRE-bGH; 2.0 × 10^13; Iontophoresis, 5 min; Addgene); Cre-dependent AAV-FLEX-RFP (AAV1-CAG-Flex-tdTomato-WPRE-bGH; 1.1 × 10^13; Iontophoresis, 5 min; Addgene).

For Cre-dependent TVA receptor-mediated rabies tracing (or TRIO tracing), the tracers used included AAVretro-Cre (AAVretro-EF1a-Cre; 2.4 × 10^13; Iontophoresis, 5 min; Salk Institute); AAV1-hSyn-FLEX-TVA-2A-GFP-2A-G (2.3 × 10^13; Pressure, 80 nl; Addgene); and EnvA G-deleted rabies dsRedXpress (2.26 × 10^10; Pressure, 40 nl; Ian Wickersham lab, MIT).

To determine synaptic circuitry in MORF3 mice, the tracers AAVretro-Cre (AAVretro-EF1a-Cre; 2.4 × 10^13; Iontophoresis, 5 min; Salk Institute) and AAV1-hSyn-DIO-mGFP-2A-SypRuby-WPRE (2.7 × 10^13; Pressure, 100 nl: Addgene) were used.

Finally, to unlock MORF expression in MORF3 mice to reveal the morphological details of ATN neurons, the main tracer used was AAVretro-Cre (AAVretro-EF1a-Cre; 2.4 × 10^13; Iontophoresis, 5 min; Salk Institute).

### Tracing strategies

**Cre-dependent anterograde.** To validate the exclusive brain-wide output of the AMd domains, a Cre-dependent anterograde tracing method was applied. One case ($n = 1$) was used per AM domain. In brief, AAVretro-Cre was injected into a downstream target of an AMd domain (e.g., PL) to deliver Cre to the AMd (in this case, AMd.medial). Next, a Cre-dependent AAV expressing either GFP or tdTomato was injected in the AMd. This method reveals the brain-wide output of AMd.medial→PL projecting neurons.

**Cre-dependent TVA receptor mediated rabies tracing method.** To identify mono-synaptic inputs to neuronal populations defined by projections in the AMd (e.g., brain-wide input to AMd.medial→PL neurons), we employed a Cre-dependent TVA receptor-mediated rabies tracing strategy sometimes referred to as TRIO (tracing the relationship between input and output) tracing. These cases were typically used to validate brain-wide input to the AM domains and one case ($n = 1$) was used per AM domain. For example, an AAVretro-Cre injection was made into a downstream projection target of AMd.medial (e.g., PL), which delivered Cre to AMd.medial neurons. Next, a Cre-dependent TVA- and RG-expressing helper virus (AAV8-hSyn-FLEX-TVA-P2A-GFP-2A-oG) and an mCherry-expressing EnvA G-deleted rabies virus were injected into the AM thereby revealing brain-wide ROIs→AMd.medial→PL connections.

### Tissue processing and imaging in 2D

Tracer transport was allowed for either one (PHAL, CTB, FG, RVΔG) or 3 (AAVs/viral tracers) weeks before animals were perfused, and their brains were extracted. For all experiments except those involving MORF3 mice to assess morphology, a 2D tissue processing workflow was employed. After administering an overdose injection of sodium pentobarbital (Euthasol, 100 mg/kg, intraperitoneal injection), each animal underwent transcardial perfusion with 50 ml of 0.9% NaCl followed by 50 ml of 4% paraformaldehyde solution (PFA; pH 9.5). Brains were post-fixed in 4% PFA for 24–48 h at 4 °C before being embedded in 3% Type I-B agarose for sectioning. Coronal sections, each 50-μm thick, were sliced into four equivalent series using a vibratome.

One of the four series of sections underwent immunostaining for the antigen of interest using the free-floating method. Sections were first transferred to a blocking solution containing normal donkey serum and Triton X-100 for 1 h. After three 5-min rinses, sections were incubated in a KPBS solution with donkey serum, Triton, and the appropriate antibody [1:1000 rabbit anti-PHAL antibody (Vector Laboratories, #AS-2300), 1:4000 mouse anti-Cre recombinase antibody (EMD Millipore, #MAB3120), or 1:500 rabbit anti-tryptophan hydroxylase (TPH2) to identify serotonin positive neurons

(ThermoFisher Scientific, #PA1-778] for 48–72 h at 4 °C. Following three KPBS rinses, sections were soaked for 3 h in the secondary antibody solution, which contained donkey serum, Triton, and a 1:500 concentration of anti-rabbit IgG conjugated with Alexa Fluor® 488 or 647 (Invitrogen, 488: #A-21206; 647: #A-31573) for PHAL and serotonin staining. For Cre recombinase staining, the secondary solution contained donkey serum, Triton, and a 1:500 concentration of anti-mouse IgG conjugated with Alexa Fluor® 488 or 647 (Life Technology, 488: #A-21202; 647: #A-31571). After three KBS rinses, the sections were counterstained with a fluorescent Nissl stain, NeuroTrace® 435/455 (NT; 1:500; Invitrogen, #N21479), mounted, and coverslipped using 65% glycerol. Finally, the sections were scanned at 10x magnification as high-resolution, multichannel virtual slide image (VSI) files using an Olympus VS120, with identical exposure parameters maintained across all cases. Pseudo coloring, artifact removal, and changes in brightness and contrast were applied to images presented in the paper. These post-imaging modifications were made to optimize visualization of tracer labels for the figures, and do not affect the conclusions of the experiments (see "Data reproducibility" section below).

### Data reproducibility

Reported connections underwent validation through at least one of the following methods. Injections targeting different regions were repeated (Supplementary Table 1) to evaluate consistency of labels. The Cre-dependent anterograde and TVA receptor-mediated rabies also validated the connections. Further, retrograde tracers were introduced into regions displaying anterograde terminal labeling to confirm anterograde connections, whereas anterograde tracers were administered into the sites of retrogradely-labeled projection cells to validate retrograde injection data. Single-neuron tracing experiments provided additional validation of select connections.

Although representative images were used in all figures, multiple injections were made in all the ROIs for reproducibility. These are clearly listed in Supplementary Table 1. For example, we made 14 injections in the ACA, 12 in ENTl, 10 in the ILA, 11 in the PL, and so on. Some of these repeated injections are presented in Supplementary figures. In Supplementary Fig. 3b, repeated injections in the rostral and caudal ACA clearly show consistent labeling in the AMd.dorsomedial and AMd.dorsal, respectively. Supplementary Fig. 3d shows repeated injections in the RSPv consistently labeling the AMd.dorsolateral domain. In Supplementary Fig. 3e, repeated injections in the PTLp (caudal, medial) show consistent labels in the AMd.lateral, while an injection in the PTLp (caudal, lateral) does not. An $n = 1$ was used only for Cre-dependent anterograde and for TVA receptor mediated rabies tracing experiments, which were used to validate the domain specific connections revealed by chemical tracers like PHAL, CTB, and FG.

### 2D post-image processing workflow

To accurately compare connectivity tracer signals across experiments, we utilized our high-throughput image processing workflow, Outspector, to achieve (1) high-quality image registration, (2) tracer signal segmentation, and (3) tracer signal quantification. Outspector provides a user-friendly interactive interface and deploys all processes on a high-performance computational infrastructure that allows high-throughput processing of large TB-sized datasets. This post-image processing Outspector pipeline is summarized as follows: each 50 μm-thick section was initially matched and subsequently aligned (registered) to its corresponding atlas level in the Allen Reference Atlas (ARA; available at http://mouse.brain-map.org/static/atlas) (Fig. 1a–c). Our semi-automated registration pipeline applies a diffeomorphic registration approach, allowing iterative modifications based on user feedback, enhancing the accuracy of registered images. Next, threshold parameters were individually adjusted for each case and tracer signal. Conspicuous artifacts in the threshold output were filtered. The

final overlap processing step generates a file with annotated (quantified) values: pixel density for anterograde tracers and cell counts for retrograde tracers.

## 2D data analysis

**Identifying AD, AV, and AM domains and their relative boundaries.** Our pipeline conducted connectivity analysis for the AV, AD, and AM. After segmenting tracer labeling, grid-based overlap annotation was performed in which the AD, AV, and AM were divided into $105 \times 105$ pixel square grids (equivalent to $63\,\mu m^2$) to tabulate labeling within distinct domains of the ATN nuclei. This was done for ARA atlas levels 61 and 63 for the AD and ARA level 61 for the AV and AM. In the given analysis, anterograde and retrograde were combined. The case specific annotations were then aggregated into a single matrix and Louvain community detection was conducted with gamma 0.75. Grid cells were color-coded according to community assignment and reordered such that the resulting Louvain clusters were placed along the diagonal of the visualized matrix (Fig. 1b–d).

**Identifying brain-wide input/output connectivity of AM domains.** To assess domain-specific AM connections, Cre-dependent anterograde and retrograde tracing strategies were applied utilizing connectionally guided delivery of Cre. For each case, sections spanning the entire brain were registered (warped), and tracer labels were segmented and annotated based on ROI. The annotated data (pixel intensity for anterograde, cell counts for retrograde) were subsequently normalized. Neuroanatomical tracing experiments across the entire brain entail numerous sources of variation, including tracer injection volume, quality of the collected tissue, variations in the plane of sectioning and immunostaining quality, accuracy of tissue registration to the atlas template, as well as microscopic imaging settings, among others. Conducting meaningful inter-experimental data comparisons necessitates the standardization of these experimental variations.

Given that tissue sections within a single specimen typically experience consistent experimental conditions, we utilize a self-normalization method to derive whole-brain connectivity fractions for each brain specimen. The connectivity fraction characterizes the proportion of connectivity found in a region out of all connectivity found in the entire specimen. The fraction calculation performs a total intensity normalization within each brain specimen, effectively normalizing away the effects of experimental condition variations. The fraction calculation was as follows: $f$ = Count/Sum of all ROI count across the case, where count is the number of labeled pixels (pixel count for anterograde, cell count for retrograde).

We also define a connectivity density vector $d$ that approximates the density of connectivity at each gray matter region. The density vector is derived from the fraction vector and represents the projection densities at each brain region of each specimen. It maintains the total counts of segmented pixels (for anterograde tracers) or cells (for retrograde tracers) constant across all specimens. This density measure offers an intuitive understanding of the connectivity data, as connectivity densities are commonly discussed in neuroanatomical literature, are easily understood, and considers ROI sizes. The density calculation was as follows: $d$ = (Fraction/area)/maximum density value.

Density values were used to generate the bar graphs depicting the connectivity of AM domains (e.g., Fig. 6h, j). They were also used to perform hierarchical 2D clustering of the AM domain-specific tracing data to determine and visualize the similarity of whole-brain projection patterns of the different injections (e.g., Fig. 1f).

## 3D tissue fixation, clearing, staining, and imaging pipeline

Following an overdose injection of sodium pentobarbital (Euthasol, 100 mg/kg, intraperitoneal injection) and 4% PFA perfusion, 250 μm-thick (for synaptic connectivity experiments) and 800 μm-thick (for

morphological assessments) tissue samples were drop fixed in 4% PFA at 4 °C for up to 2 weeks, then transferred into PBSN (1× PBS + 0.02% sodium azide and stored at 4 °C until SHIELD epoxy fixation. For SHIELD epoxy fixation, samples were incubated in the SHIELD OFF solution at 4 °C for 3 days with agitation, then transferred into pre-heated SHIELD ON solution and incubated at 37 °C overnight with agitation. After the SHIELD ON step, samples were either transferred into PBSN and stored at 4 °C until the delipidation step or washed 2× for 2 h and moved onto the delipidation step. When ready for delipidation, samples were put into 10 mL LifeCanvas's Active delipidation buffer and incubated overnight at 45 °C with agitation. The next day, samples began active delipidation using LifeCanvas's SmartBatch+ pre-installed active clearing protocol (40 V, 30 h). Following active delipidation, samples were washed 2 × 2 h with 1× PBSN and stored in 1× PBSN at 4 °C until ready for staining. When samples were ready for staining, they were incubated in a 10 mL primary sample buffer with 5% NDS overnight at 37 °C. Replacing this solution once in the morning, samples were then put in the SmartBatch+ to begin the active primary V5 staining protocol [1:1000 anti-V5, Fortis, #A190-119A (goat) or #A190-120A (rabbit)]. Sample solution contained primary antibody added to the primary sample buffer + 5% NDS, and LifeCanvas's primary conduction buffer was used (18–22 h). Primary staining concluded when the sample buffer reached <pH 8.0. Following the primary stain, samples were washed 2 × 2 h in 1× PBSN then incubated in 4% PFA overnight at 4 °C to prevent primary antibody dissociation. After 4% PFA incubation, samples were washed in 1× PBSN at RT for 2 h, then 10 mL secondary sample buffer + 5% NDS at 37 °C for 2 h with agitation. Samples were then put in the SmartBatch+ for one 2-h active secondary sample buffer + 5% NDS wash using the active secondary protocol pre-installed in the SmartBatch+. Active secondary protocol was run with secondary antibodies (1:1000 in-house conjugated anti-goat and anti-rabbit Fab-Setau-647; see following section for details regarding secondary antibody conjugation procedure) in the secondary sample buffer + 5% NDS for 12 h. The next morning, samples underwent 2 × 2 h active washes with only the secondary sample buffer. Samples were then washed 2 × 2 h in 1× PBSN and then incubated in 4% PFA overnight at 4 °C to fix secondary antibodies in place. The next morning, samples were washed 2 × 2 h in 1× PBSN and then were either placed into storage in 1× PBSN at 4 °C or moved onto the light sheet or spinning disc confocal preparation.

Tissue being processed for the LifeCanvas SmartSPIM light sheet microscope at 15 × (for morphological assessment) were moved into 10 mL delipidation buffer containing 1:500 of a nucleic acid stain [Syto13 (ThermoFisher Scientific, #S7575) or Propidium Iodide (ThermoFisher Scientific, #P1304MP)] and incubated overnight at 37 °C with agitation. The following morning, samples were washed 2 × 2 h in 1× PBSN then incubated in 4% PFA at 4 °C overnight. The next morning, samples were washed 2 × 2 h in 1× PBSN then either stored in 1× PBSN at 4 °C, or immediately moved into 50% EasyIndex and incubated overnight at 37 °C then put into 100% EasyIndex and incubated overnight at 37 °C. The next morning, samples were mounted onto a sample holder using 1.5% agarose and superglue and were put into the light sheet chamber for imaging.

Tissues processed for Olympus Dragonfly spinning disk confocal imaging were sectioned at either 800 μm for 20 × imaging (for morphological assessment) or 250 μm for 60 × imaging (for synaptic connectivity experiments). Sections were then incubated in 1:500 DAPI (ThermoFisher Scientific, #D1306) in 1× PBSN solution or immediately moved into 50% EasyIndex and incubated overnight at RT. The next day samples were then put into 100% EasyIndex and incubated for at least 3 h or were held in a parafilm-wrapped 24-well plate and protected from light until ready for slide mounting. Sections were mounted onto plain microscope slides with Sunjin spacers and coverslipped with 1.5H coverslips that were held in place with superglue.

### SeTau-647 secondary antibody conjugation

For optimal signal retention of MORF3 V5-tagged neurons in 3D imaging, we chose to conjugate SeTau-647-NHS ester (Setabiomedicals, #K9-4149) to AffiniPure Fab fragments (Fab donkey anti-rabbit, Jackson Immunoresearch, #705-007-003; Fab donkey anti-goat, Jackson Immunoresearch, #711-007-003) because of the intensity of emission and its resilience to photobleaching. Fab's were conjugated to SeTau-647-NHS ester using a 1:2 molar ratio (Fab:Dye). 1 M sodium bicarbonate, at 10% of the volume of the Fab:Dye solution, was added to the Fab:Dye solution (e.g., if Fab:Dye solution is 100 μl, then add 10 μl of sodium bicarbonate for a final solution volume of 110 μl) and was agitated at 500 rpms at room temperature for 1 h, then excess dye was removed using size exclusion columns (Zeba™ spin desalting columns and plates, 40K MWCO, 0.5 mL, ThermoFisher Scientific, #A57760).

### 3D neuron reconstruction and mapping

To accurately select the neurons from the regions of interest (i.e., AD, AV, and AM domains), the Orthoslicer and FilamentTracer tools within Imaris (BITPLANE, RRID:SCR_007370) were used to manually identify the (somatic) location of each neuron. The Orthoslicer thickness of the tissue was adjusted to be between 10 and 30 μm. The cell body of each neuron was ensured to be within the selected region's border in each of the coronal views of the tissue, located between ARA 58-66 on the Allen Reference Atlas. The identified neurons were scaled from voxel to micron dimensions (and scaled back to voxel dimensions if required) using a custom Python script and were manually reconstructed within the Terafly program in Vaa3d[93]. For precise quality check of the reconstructions, the neurons were opened and edited in neuTube[94] to ensure correct branch typing, location of branch points and proper connections of the nodes. Each digital SWC reconstruction is a tree structure constituting a series of connected frustums/compartments, where a compartment is represented by a single row (containing seven columns) of an SWC file[95]. The compartment's id, type (dendrite, axon, or soma), end point coordinates (X, Y, Z location), and the end point thickness are represented by the first six columns. The 7th column of the SWC row has the id of the compartment's origin point/parent compartment (for additional details, see swc-specification.readthedocs.io). Imaris was once more used to visualize and further quality check the neuron morphology and the location markers, such as identifying neurons lying closer to the borders of the regions of interest (edge) versus the ones that are more within the center of the AM (non-edge). A total of 109 neural reconstructions were analyzed for this study, which included 10 AD, 82 AM, and 17 AV neurons. Out of the 82 AM neurons, 72 neurons were from the AMd region, and the remaining 10 were from the AMv region. The AMd neurons were subdivided into five domains: (i) AMd.core (12 neurons), (ii) AMd.dorsal (12 neurons combined from AMd.dorsal and AMd.dorsolateral domains), (iii) AMd.dorsomedial (11 neurons), (iv) AMd.lateral (20 neurons), and (v) AMd.medial (17 neurons from both the AMd.medial tip and AMd.ventromedial domains). Some domains were combined for this analysis, given the low number of reconstructed neurons in each of the 7 AMd domains.

To reconstruct the full 3D morphology of neurons, including both dendrites and axons (Fig. 7m), we utilized Vaa3D (http://vaa3d.org[95–97] along with its recent successor, Collaborative Augmented Reconstruction (CAR, https://github.com/neurogeom/CAR[98]. This approach was applied to fMOST images obtained through the collaboration[99]. To guarantee the precision of the neuron locations and morphologies, we conducted human inspection of the somas and their 3D structures.

### Statistical analyses of morphometrics

We first quantified basic morphometric features that demonstrated distinct and complementary aspects of dendritic architecture. These features included total wiring (total dendritic length), total complexity (number of dendritic branches), arbor height, maximum path distance, and average branch order.

We noticed distinct morphological features for AMd neurons whose somas were on the border of the AMd (edge, $n = 27$) compared to those whose somas were located more internally (non-edge, $n = 45$). To investigate this, we performed quantification of two attributes for these edge or non-edge AMd neurons, specifically to measure the orientation bias (towards the AMd center) of a neuron's overall dendritic architecture. The first feature is the average angular deviation relative to the local core vector (Fig. 10). The deviation of every dendritic compartment vector relative to its local core vector (where the local core vector of a compartment is a straight line connecting the compartment's origin point to the AMd center) is averaged across all compartments for a collective average deviation for that neuron. An acute compartment deviation (deviation of less than 90°) would mean that the dendritic compartment's endpoint is closer to the AMd center than its origin point. An obtuse compartment deviation, on the other hand, would mean that the compartment's endpoint is further away from the center compared to the compartment's origin point. Neuronal nodes are resampled for this measurement so that each compartment (i.e., internode distance) is ~1 μm in length. Therefore, a neuron with a lower average angular deviation (from the AMd center) would have greater tropism/bias towards the AMd center. The second feature measured the proportion of total dendrite that lay closer to the AMd core than the neuron's cell body (i.e., proportion of dendrites whose distance from the core was lower than the distance between the core and that neuron's cell body). Hence, neurons with a higher proportion of dendrites closer to the center also demonstrate greater tropism/bias towards the AMd center.

We carried out a comparative morphometric study on three levels of groupings due to smaller sample sizes for both AD ($n = 10$) vs AM ($n = 82$) vs AV ($n = 17$) comparison, as well as for the AMd domain comparisons. The domains included the AMd.core ($n = 12$), AMd.dorsal (AMd.dorsal and AMd.dorsolateral combined, $n = 12$), AMd.dorsomedial ($n = 11$), AMd.lateral ($n = 20$), and AMd.medial (AMd.medial tip and AMd.ventromedial combined, $n = 17$). Because of smaller sample sizes, pairwise two-sided Wilcoxon rank sum tests were used for each pair (three pairs in AD vs. AM vs. AV comparison and a total of 10 pairs for the five AMd domains). For the AMd edge vs non-edge analysis, a two-sided t-test was carried out since the sample size of both groups were higher and the distributions were normal. False discovery rate (FDR) corrections were carried out for all multiple comparisons.

### Synapse reconstructions

Soma, dendrites, and spines were reconstructed using the Soma, Tree, and Spine module respectively in Neurolucida 360 software, while axon boutons that are adjacent to the soma, dendrites, and spines were reconstructed with the Puncta module. Briefly, soma was created using the default setting, and dendrites were created using the user-guided mode with the Rayburst Crawl mode. Spines were then automatically segmented with the following detection settings: outer range set to 5 μm, minimum height set to 0.3 μm, detector sensitivity set to 120%, and minimum count set to 10 voxels. Axons were created using the machine learning method with the maximum distance to soma, dendrites, or spines set to 1 μm. Data analysis was performed in Neurolucida Explorer software. For quantifying the number of synapses, the overlap percent to colocalize was set to 10%.

### Reporting summary

Further information on research design is available in the Nature Portfolio Reporting Summary linked to this article.

## Data availability

Source data are provided in a Source Data file. Source data are provided with this paper.

## Code availability

The code used to analyze the data for this project can be found here: https://github.com/ucla-brain/atn_community.

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

## Acknowledgements

Funding for this project was provided by NIH Grants U01MH114829 (H.-W.D.) and 1R01NS133744-01 (H.-W. D.).

## Author contributions

H.H., H-W.D. conceived and designed the project. H.H. managed the project and conducted the data analysis, H.H., H.-W.D. wrote the manuscript, and prepared the figures for publication. M.R. conducted all of the sectioning, staining, clearing, and imaging of thick tissue, the staining and annotation of serotonin labels, and the manual mapping of the AM labeling. S.N. mapped registered axonal projections onto the Allen Brain Common Coordinate Framework (CCF) and calculated axonal projections to individual brain regions. S.N. developed analysis code, performed all dendritic morphological analyses, generated all the corresponding graphs, and wrote the corresponding portions of the paper. A.G., H.-S.M., Q.X. carried out the dendritic reconstructions of neurons. H-S.M. mapped the reconstructed neurons onto the atlas template. L. Garcia, I.B. managed and executed the analysis of the 2D dataset (community detection, hierarchical clustering) and generated their corresponding visualizations. D.L., T.B. performed all the post-image processing for the 2D datasets in Outspector. J.S. performed the synapse reconstructions. C.E., S.Y. assisted with the online development and visualization of the wiring diagram. Y.E.H. managed the breeding of MORF3 mice. L. Gou, B.Z. performed tracer injection surgeries, while C.C., J.G., H.X., and I.Y. performed histological processing and imaging of 50 μm sections. M.Z. generated the code for hierarchical clustering. I.B., K.M., S.N., and A.D. developed code for image processing and digital morphological reconstructions. Q.Z. assisted with high resolution imaging of neurons. L. Liu, X.C., and Z.Y. performed neuron axonal reconstructions, while H.P. managed that portion of the project. N.N.F. assisted in the design of experiments, with figure arrangements, and contributed to the editing of the manuscript.

## Competing interests

The authors declare no competing interests.
