## [Transparent Peer Review file · Nature Communications]

Distinct subnetworks of the mouse anterior thalamic nuclei

Corresponding Author: Dr Hourii Hintiryan

This manuscript has been previously reviewed at another journal. This document only contains information relating to versions considered at Nature Communications. Any other journal names are redacted.

Version 0:

Reviewer comments:

Reviewer #1

(Remarks to the Author)

The authors have addressed all my previous concerns in the letter and revised manuscript. I would recommend the publication in Nature Communications.

Reviewer #2

(Remarks to the Author)

This revised anatomy paper by Hintiryan et al. examines cells and connections in the mouse anterior thalamic nuclei (ATN). The ATN consists of anterodorsal (AD), anteroventral (AV), and anteromedial (AM) nuclei, which link multiple other brain regions. The ATN is vital for a variety of behaviors ranging from learning and memory to navigation and motivation, and important to characterize. Here the main approach is to use anterograde and retrograde tracers with automated and manual image analysis to systematically map a variety of pathways through ATN. The resulting anatomy identifies both general connectivity (Figure 1), as well as a vast array of subnetworks, including in AD (Figure 2 & 3), AV (Figure 4), and AM (Figure 5 & 6). Cells in these nuclei are then further analyzed to describe dendritic morphology (Figure 7) and more fine-grained connectivity (Figure 8). The revised paper again provides a dizzying amount of information about the cells in ATN and the specificity of inputs and outputs. I agree with the authors that this large data set may help resolve some inconsistencies in the literature about multiple parts of ATN. I also agree that while the data does not identify genetic cell types, the anatomy alone represents a potentially useful advance. While there is no completely new technology presented in the paper, I also think that is not an issue for this particular manuscript. However, for me the main problem of the paper continues to be that it is confusing organized and presented, often with mismatched text and figures that make it almost indecipherable for any but the most dedicated reader. While I think it could be ultimately suitable for publication, the paper needs to be reorganized and rewritten to make it much more accessible.

New Specific Comments:

The main issue with the paper is the confusing organization and presentation. There is a lot of interesting data, but the text jumps from figure to figure, and panel to panel, in a haphazard way. Below I list some ways that might improve the paper, but it will be a nontrivial task to reorganize this work...

1. Figure 1: Fix so that panels read from left to right (e.g., B should be a row, E should be to the left of F). Also fix the order in the text (e.g., E is currently referred to after F). The authors could also consider splitting A-D and E-H, which might allow them to provide more examples and details.
2. Figure 1G&H: The injection sites should be shown in the figures. It is hard to interpret the data otherwise.
3. Figure 2: Standardize nomenclature (ADm or AD.medial; ADl or AD.lateral); do not switch back and forth.
4. Figures 2 & 3 are hard to follow. The results are mostly qualitative, and presented as observations, which is expected for this type of work. However, the text refers to panels out of order, jumps from Figure 2 to Figure 3, and eventually returns to

Figure 2. The authors should present the figures and text in a logical, linear order, and try to make it as accessible as possible to the reader.

5. Figure 4 is also often out of order, which again makes it very hard to follow, interpret, and assess.

6. Figures 5 & 6 have the same issue. For example, after Figure 5a-c comes Figure 6, leaving out Figure 5d-n. When the text returns to Figure 5, it seems to skip many panels. Again, the data is interesting, but inaccessible.

7. Figure 7: The use of MORF3 mice needs to be motivated. The authors need to explain why this analysis was only done for these two connections. There should also be some quantification of putative connections.

8. Figure 8: The morphology analysis is interesting but should be better motivated. While the authors swapped the order of Figures 7 & 8 to match the text, it would have been better to alter the text to match the figures.

Specific Comments (continued from previous review):

1. The new discussion ("Caveats to tracing experiments") is helpful, but it would be better to include more of this information in the Results, which will certainly be on the minds of most readers.

2. The addition of technical details to the methods is welcome, but it would be helpful if more mention was made in the Results. Many of the scale bars are still completely unreadable.

3. The explanation of the N is helpful, but they are still very variable (ranging from N = 1 to 24). The authors explain why the low N sometimes do not matter, but it would be best if they were at least N = 3.

4. Swapping Figures 7 and 8 is helpful, but the order of the figures and text is still extremely confusing.

Reviewer #3

(Remarks to the Author)

I know this paper was transferred from [journal name redacted]. In this manuscript, the authors applied genetic sparse labeling, brain clearing, 3D microscopic imaging, and computational informatics to classify neuron types of the entire ATN, determined their brain-wide connectivity profile at the single neuron and synaptic resolutions, and characterized their morphological features. This is the first systematic and extensive examination of ATN connectivity that can reveal the exceedingly granular subnetworks to account for ATN functional diversity. I really appreciate that the authors added 'Caveats to tracing experiments' in the discussion section. Answered some of my questions during the first review. But there still a few comments are listed as below.

1. In Fig 1g, some labeled areas are making a footnote, while others are not(2 pictures on the right).

2. In Fig 2a and Fig 2c, retrograde tracer CTB injected in the PRE, its input brain area AD is marked with inconsistent shapes. Is this due to inconsistent labeling at the injection site between the two experiments? Or the connection between PRE and AD is not a relatively fixed area?

3. In Fig 2k, the green fluorescence corresponding to the CTB in the middle looks different from that on the right side. Is the exposure time different during imaging?

4. Fig 2m, Fig 3f, Fig 5f, Fig 6c and so on, when calculating the projection or input intensity of a certain brain region, please specify the number of animals or brain slices involved in this statistical analysis. The number of test animals must be greater than 2 to eliminate differences caused by different individuals.

5. To analyze the general ATN connections, AD, AV and AM subnetworks, the author used different anterograde and retrograde tracers in this work. Sun L etc. had found that different multi-trans-synaptic and mono-synaptic retrograde viral tracers exhibited discrepant neurotropism within certain brain regions (PMID: 30736827). So, it is necessary to appropriately analyze the possible impact of the use of different tracers on the results in the discussion section of the paper.

Version 1:

Reviewer comments:

Reviewer #2

(Remarks to the Author)

This revised manuscript by Hintiryan et al. examines in detail multiple neural circuits involving mouse anterior thalamic nuclei. The paper authors have responded well to the previous critiques, extensively reorganizing both the figures and text, and the paper is enormously improved. I have only minor comments, and the authors should be congratulated on an impressive study, which is ready for publication.

Minor comments:

1. Most of the figures are much better labeled, but there is still some room for improvement. The mentions of Figure 7 start to

go out of order, but should be easy to fix. Some of Figure 8 is also out of order, and Figure 8c-d do not appear to be mentioned in the text. Figure 9 is mislabeled in several cases, incorrectly referred to as Figure 8. Fixing these issues will make the text even easier to read.

2. I still think the scale bars are almost completely unreadable in many cases. It is true that they can be zoomed in, but most readers are not going to do that. I would suggest either making them readable in standard view, or putting the numbers in the figure legends.

Reviewer #3

(Remarks to the Author)

The authors have addressed all my previous concerns in the letter and revised manuscript.

Point by point response to reviewers

Reviewer #1 (Remarks to the Author):

The authors have addressed all my previous concerns in the letter and revised manuscript. I would recommend the publication in Communications.

Response: Thank you.

Reviewer #2 (Remarks to the Author):

This revised anatomy paper by Hintiryan et al. examines cells and connections in the mouse anterior thalamic nuclei (ATN). The ATN consists of anterodorsal (AD), anteroventral (AV), and anteromedial (AM) nuclei, which link multiple other brain regions. The ATN is vital for a variety of behaviors ranging from learning and memory to navigation and motivation, and important to characterize. Here the main approach is to use anterograde and retrograde tracers with automated and manual image analysis to systematically map a variety of pathways through ATN. The resulting anatomy identifies both general connectivity (Figure 1), as well as a vast array of subnetworks, including in AD (Figure 2 & 3), AV (Figure 4), and AM (Figure 5 & 6). Cells in these nuclei are then further analyzed to describe dendritic morphology (Figure 7) and more fine-grained connectivity (Figure 8). The revised paper again provides a dizzying amount of information about the cells in ATN and the specificity of inputs and outputs. I agree with the authors that this large data set may help resolve some inconsistencies in the literature about multiple parts of ATN. I also agree that while the data does not identify genetic cell types, the anatomy alone represents a potentially useful advance. While there is no completely new technology presented in the paper, I also think that is not an issue for this particular manuscript.

Response: Thank you.

However, for me the main problem of the paper continues to be that it is confusing organized and presented, often with mismatched text and figures that make it almost indecipherable for any but the most dedicated reader. While I think it could be ultimately suitable for publication, the paper needs to be reorganized and rewritten to make it much more accessible.

Response: We extensively simplified and reorganized the figures and correspondingly revised the text to significantly improve its readability and clarity. The text now follows the order of the new figures, making it easier to read and understand. Additionally, thank you for your specific comments regarding the figure panels and corresponding explanations as to why they would be a source of confusion for the reader. This was very helpful since sometimes these details are a bit challenging to pinpoint when one is entirely entrenched in the details of a manuscript.

New Specific Comments:

The main issue with the paper is the confusing organization and presentation. There is a lot of interesting data, but the text jumps from figure to figure, and panel to panel, in a haphazard way. Below I list some ways that might improve the paper, but it will be a nontrivial task to reorganize this work...

1. Figure 1: Fix so that panels read from left to right (e.g., B should be a row, E should be to the left of F). Also fix the order in the text (e.g., E is currently referred to after F). The authors could also consider splitting A-D and E-H, which might allow them to provide more examples and details.

Response: Figure 1 was reorganized so that the panels are now much easier to follow.

2. Figure 1G&H: The injection sites should be shown in the figures. It is hard to interpret the data otherwise.

Response: Including the injections sites for Figure 1f and 1g is a reasonable suggestion. Given the multiple injections per brain, it would really crowd the figure to include all the injections in this panel. However, the labeling patterns presented in Figure 1g and h are explored in greater detail in subsequent figures, which include all of the injection sites. For example, for the AAV SUBd injection, the injection site is in Extended Data Fig. 2g. The FG CP injection, the injection site is in Extended Data Fig. 6c. Further, in subsequent figures, for all AV, AD, and AM injections we have included the labeling in mammillary bodies as a verification for their injection sites (e.g., Fig. 3b). We could include this information in the figure legend, but this will require the reader to jump to out of order Extended Data Figures.

3. Figure 2: Standardize nomenclature (ADm or AD.medial; ADl or AD.lateral); do not switch back and forth.

Response: This is correct. We have the full domain name like AD.medial and it's abbreviation, ADm. Because there are so many domains, it would be difficult for the reader to track the abbreviations. Therefore, we use the full name within the text, which minimizes the effort required by the reader to remember what it stands for, but the abbreviations are sometimes used in the figures due to space limitations. The nomenclature should all be consistent throughout the updated text and figures.

4. Figures 2 & 3 are hard to follow. The results are mostly qualitative, and presented as observations, which is expected for this type of work. However, the text refers to panels out of order, jumps from Figure 2 to Figure 3, and eventually returns to Figure 2. The authors should present the figures and text in a logical, linear order, and try to make it as accessible as possible to the reader.

Response: We have reorganized original Figures 2 and 3, and updated the text to ensure that the figure panels follow the order presented in the text. The new Figure 2 now focuses on AD connections. Figure 3 highlights the AV connections, while Figures 4-7 address domains of the AM. Figure 8 illustrates the synaptic connections of the AM. The new Figure 9 focuses on the ATN's connections with the mammillary body and brainstem structures. With these changes, all figure panels are now consistently arranged to align with the order described in the text.

5. Figure 4 is also often out of order, which again makes it very hard to follow, interpret, and assess.

Response: With the extensive reorganization of the figures, Figures 4-7 now illustrate the connectivity of the AMd and AMv domains. Consequently, the figures should be clearer and easier to follow within the text.

6. Figures 5 & 6 have the same issue. For example, after Figure 5a-c comes Figure 6, leaving out Figure 5d-n. When the text returns to Figure 5, it seems to skip many panels. Again, the data is interesting, but inaccessible.

Response: The reorganization of the text and the figures has now resolved this issue.

7. Figure 7: The use of MORF3 mice needs to be motivated. The authors need to explain why this analysis was only done for these two connections. There should also be some quantification of putative connections.

Response: Traditionally, synaptic connections have been validated primarily through electron microscopy (EM), which provides a definitive ground truth for these connections. An alternative method involves the combination of single-cell slice recording or patch-clamp techniques with optogenetic stimulation. However, these methods are not suitable for broadly mapping many synaptic connections due to their limited scope and scalability. For broader application, classical tract tracing methods such as anterograde tracing can label axonal terminals originating from upstream structures, while retrograde tracing can identify postsynaptic neurons. However, traditional tracers like Fluoro-Gold (FG) or cholera toxin subunit B (CTB) primarily label neuronal somas and often miss critical postsynaptic structures like dendrites. Immunostaining methods that could reveal dendrites are technically challenging and not always effective.

In contrast, the MORF3 labeling technique offers a significant advancement by clearly delineating dendritic arbors and spines, encompassing the entire spectrum of postsynaptic elements. This enhancement greatly increases the likelihood of accurately surveying synaptic connections. Specifically, thalamic neurons, known for their extensive and intricate branching, are well-suited for analysis using MORF3 labeling, providing an excellent tool for examining synaptic inputs to these neurons. We implemented this approach as a proof-of-principle demonstration to examine the synaptic connections of the AMd.m domains. This provides a step-by-step guide for readers interested in exploring synaptic connections in other structures. Due to space constraints, we were unable to test all potential synaptic connections.

8. Figure 8: The morphology analysis is interesting but **should be better motivated**. While the authors swapped the order of Figures 7 & 8 to match the text, it would have been better to alter the text to match the figures.

Response: Dendrites are crucial in how neurons receive and integrate signals from other cells. The shape and structure of dendrites influence signal processing and determine connectivity patterns within neural circuits. Additionally, the extensive scope of dendritic arbors defines the size of a neuron's receptive field. Dendritic morphology is also a key characteristic that distinguishes different neuronal types. However, the dendritic morphology of ATN neurons is particularly scarce in the literature. Therefore, we leveraged MORF3 labeling to address this knowledge gap. We included this more detailed explanation in the Results subsection titled *ATN dendritic morphology*.

Interestingly, the organization of neurons in vertebrates is distinct from that in invertebrates. In the vertebrate brain, there are two primary types of gray matter structures: Layered and Nuclear. The organization of dendritic arbors has been well categorized in layered structures such as the cortex and hippocampus. However, the organization is less clear in nuclear structures such as the ATN.

During our initial qualitative observations of microscopic images, we noticed that in the ATN, the organization of the edge neurons somewhat mirrored that of invertebrate (specifically *Drosophila*) neuronal organization, where the cell bodies lie on the surface of the neuropils. This observation motivated us to quantify this dendritic organization by creating a new metric for the orientation of dendritic arbors relative to the center of the nuclear structure. We observed that the dendritic orientation of "edge" neurons was analogous to that of invertebrate neurons.

The similarity of organization of dendritic arbors between edge neurons and *Drosophila* neuropile neurons suggests a conserved neural circuit architectures across evolutionary scales. This also further highlights the need to delineate the differences in neural signal pathways that exist between layered and nuclear structures. Once again, due to space limitations, we left this out of the discussion although in the current manuscript we thoroughly discuss the relevance of examining neuronal morphology.

Specific Comments (continued from previous review):

1. The new discussion ("Caveats to tracing experiments") is helpful, but it would be better to include more of this information in the Results, which will certainly be on the minds of most readers.

Response: We tried a draft of the manuscript where we included this in the Results. Ultimately, we kept this information in the discussion section and referenced it in the results section. This allowed us to focus on the more detailed scientific results without disruption of the information.

2. The addition of technical details to the methods is welcome, but it would be helpful if more mention was made in the Results. Many of the scale bars are still completely unreadable.

Response: All the scale bars in our high-resolution versions of the figures are readable. When the figure is converted to a pdf, they become less clear. But we did update the scale bars again and they should all be readable when zoomed in even in the pdf. Also, again we tested a draft of the manuscript in which we included this in the results, but ultimately the flow of the paper was hampered. Keeping those details in

the Methods allows us to focus more on the results (which are already detail heavy) without too much disruption.

3. The explanation of the N is helpful, but they are still very variable (ranging from N = 1 to 24). The authors explain why the low N sometimes do not matter, but it would be best if they were at least N = 3.

Response: Agreed, and more animals should be added if further validation is needed. Since all the data reported are validated in several ways, adding additional animals for further validation does not seem warranted here.

4. Swapping Figures 7 and 8 is helpful, but the order of the figures and text is still extremely confusing.

Response: This issue should now be corrected given the reorganization of all the figures and the text.

Reviewer #3 (Remarks to the Author):

I know this paper was transferred from [journal name redacted]. In this manuscript, the authors applied genetic sparse labeling, brain clearing, 3D microscopic imaging, and computational informatics to classify neuron types of the entire ATN, determined their brain-wide connectivity profile at the single neuron and synaptic resolutions, and characterized their morphological features. This is the first systematic and extensive examination of ATN connectivity that can reveal the exceedingly granular subnetworks to account for ATN functional diversity. I really appreciate that the authors added 'Caveats to tracing experiments' in the discussion section. Answered some of my questions during the first review.

Response: Thank you.

But there still a few comments are listed as below.

1. In Fig 1g, some labeled areas are making a footnote, while others are not (2 pictures on the right).

Response: Good catch. This is now corrected.

2. In Fig 2a and Fig 2c, retrograde tracer CTB injected in the PRE, its input brain area AD is marked with inconsistent shapes. Is this due to inconsistent labeling at the injection site between the two experiments? Or the connection between PRE and AD is not a relatively fixed area?

Response: Good question. The connection between the AD and PRE is generally in the medial AD (new Figure 2 panels b, c, and f). The slight variation in the labeling that you observe across different experiments may be a result of slight variability in the injection site location, but also due to other factors like the section that is presented. Despite these variations, all our retrograde tracer injections in the PRE result in labeling in the AD medial. This underscores the importance of validating data.

3. In Fig 2k, the green fluorescence corresponding to the CTB in the middle looks different from that on the right side. Is the exposure time different during imaging?

Response: Consistent exposure times are used to image all experiments. The inconsistency you keenly spotted was due to slight differences in the pseudo coloring that was used for the labels. This is now corrected and in Extended Data Figure 6a-b (due to the extensive reorganization of the text and figures).

4. Fig 2m, Fig 3f, Fig 5f, Fig 6c and so on, when calculating the projection or input intensity of a certain brain region, please specify the number of animals or brain slices involved in this statistical analysis. The number of test animals must be greater than 2 to eliminate differences caused by different individuals.

Response: This is correct. In this case, the charts that summarize projection or input density that are based on the Cre-dependent anterograde tracing and Cre-dependent TVA receptor mediated tracing experiments (each N=1) were only used as validation cases. Table 1 was included to show the number of injections that were made in each ROI that were used for the current paper. We used a total of 24 injections in the

AMd alone. We had 14 in the ACA, 12 in the ENTl, 10 in ILA, 11 in the PL, etc. These were the injections used to draw most conclusions and all connections were validated.

Further, the Cre-dependent tracing validation experiments were only one of many validation cases we used to show the reliability of the connections (although one validation should be sufficient). A good example of this is shown with RSPv connections with AMd.dorsolateral presented in Figure 5b. Our data show that a Phal/CTB injection in the RSPv back labels neurons only in the AMd.dorsolateral, suggesting a AMd.dorsolateral→RSPv connection. Next, we used Cre-dependent anterograde tracing to validate this connection (Figure 5c-d), and although this is an n=1, you can see in Extended Data Figure 3d, we show an additional 4 retrograde tracer injection cases in the RSPv that validate this connection. Together, the data strongly support the AMd.dorsolateral→RSPv connection.

Another example of this is provided by the PTLp (caudal, medial) connections with the AMd.lateral. We introduce this PTLp↔AMd.lateral connection in Figure 5f-g with our Phal/CTB injection in the PTLp (caudal, medial). Next, we validate the AMd.lateral→PTLp connection with Cre-dependent anterograde tracing and although this was an n of 1, this connection was also validated with 3 additional cases that show the same connection that are presented in Extended Data Figure 3e. In addition, we show that an injection in the PTLp (caudal, lateral) results in a different pattern of labeling.

The AM-ACA connections provide yet another example. In Figure 4 d-f, we introduce the specific connections of the rostral ACAv with AMd.dorsomedial and of the caudal ACAv with AMd.dorsal. We use Cre-dependent anterograde tracing to show that most connections of the AMd.dorsal are with the ACAv (Figure 4 g-h). However, in Extended data Figure 3b, we show 6 additional validation cases, 3 for the rostral ACAv and 3 for the caudal ACAv.

In the updated figure legends, we have now made it clearer that these were validation cases. This is also clearly stated in the Results section under the *Characterizing ATN domain connectivity* subsection (second paragraph) and in the Methods under the *Tracing strategies* subsection.

Also, the data were not statistically analyzed. The bar graphs were just used to visualize the data so that one can see the connections in the raw images were validated. The number of cases used for the Cre-dependent AAV and TVA receptor mediated tracing are included in the Methods.

For the matrices depicting the results of the Louvain analysis, the number of cases (and sections used) are included in the figure legends. For example, in the figure legend for Figure 2, regarding the matrix we include that “A total of 6 cases (6 sections with AD labels at ARA 61 and another 6 sections at ARA 63) were included in the analysis. Injection ROIs are indicated in the rows of the matrix”. In addition, when connectivity matrices are presented (e.g., Figure 7j, l), the number of cases used per domain (n=1) is now also specified.

5. To analyze the general ATN connections, AD, AV and AM subnetworks, the author used different anterograde and retrograde tracers in this work. **Sun L etc. had found that different multi-trans-synaptic and mono-synaptic retrograde viral tracers exhibited discrepant neurotropism within certain brain regions (PMID: 30736827).** So, it is necessary to appropriately analyze the possible impact of the use of different tracers on the results in the discussion section of the paper.

Response: Great suggestion. To address this issue along with similar or related concerns raised by the other reviewers, we have now included this in the Discussion section under the *Caveats to tracing experiments* subsection regarding the tracers used and the tropism of different viral tracers.

Response to reviewer comments

Reviewer #2 (Remarks to the Author):

This revised manuscript by Hintiryan et al. examines in detail multiple neural circuits involving mouse anterior thalamic nuclei. The paper authors have responded well to the previous critiques, extensively reorganizing both the figures and text, and the paper is enormously improved. I have only minor comments, and the authors should be congratulated on an impressive study, which is ready for publication.

Minor comments:

1. Most of the figures are much better labeled, but there is still some room for improvement.

Response: Thank you again for catching these.

The mentions of Figure 7 start to go out of order, but should be easy to fix.

Response: Corrected.

Some of Figure 8 is also out of order, and Figure 8c-d do not appear to be mentioned in the text.

Response: Corrected.

Figure 9 is mislabeled in several cases, incorrectly referred to as Figure 8. Fixing these issues will make the text even easier to read.

Response: Corrected.

2. I still think the scale bars are almost completely unreadable in many cases. It is true that they can be zoomed in, but most readers are not going to do that. I would suggest either making them readable in standard view, or putting the numbers in the figure legends.

Response: All scale bars should now be readable in standard view.